# Evaluation of Remote Sensing and Reanalysis Products for Global Soil Moisture Characteristics

**Peng Zhang** [1], **Hongbo Yu** [1,2,3,*], **Yibo Gao** [4] **and Qiaofeng Zhang** [1,2]

1. College of Geographical Science, Inner Mongolia Normal University, Hohhot 010022, China; zhangpeng@mails.imnu.edu.cn (P.Z.); zhangqf@imnu.edu.cn (Q.Z.)
2. Inner Mongolia Key Laboratory of Remote Sensing and Geographic Information Systems, Hohhot 010022, China
3. Provincial Key Laboratory of Mongolian Plateau's Climate System, Hohhot 010022, China
4. College of Tourism and Urban-Rural Planning, Chengdu University of Technology, Chengdu 610059, China; gaoyibo@stu.cdut.edu.cn
* Correspondence: 20041308@imnu.edu.cn; Tel.: +86-188-4712-3375

**Abstract:** Soil moisture (SM) exists at the land-atmosphere interface and serves as a key driving variable that affects global water balance and vegetation growth. Its importance in climate and earth system studies necessitates a comprehensive evaluation and comparison of mainstream global remote sensing/reanalysis SM products. In this study, we conducted a thorough verification of ten global remote sensing/reanalysis SM products: SMAP DCA, SMAP SCA-H, SMAP SCA-V, SMAP-IB, SMOS IC, SMOS L3, LPRM_C1, LPRM_C2, LPRM_X, and ERA5-Land. The verification was based on ground observation data from the International SM Network (ISMN), considering both static factors (such as climate zone, land cover type, and soil type) and dynamic factors (including SM, leaf area index, and land surface temperature). Our goal was to assess the accuracy and applicability of these products. We analyzed the spatial and temporal distribution characteristics of global SM and discussed the vegetation effect on SM products. Additionally, we examined the global high-frequency fluctuations in the SMAP L-VOD product, along with their correlation with the normalized difference vegetation index, leaf area index, and vegetation water content. Our findings revealed that product quality was higher in regions located in tropical and arid zones, closed shrubs, loose rocky soil, and gray soil with low soil moisture, low leaf area index, and high average land surface temperature. Among the evaluated products, SMAP-IB, SMAP DCA, SMAP SCA-H, SMAP SCA-V, and ERA5-Land consistently performed better, demonstrating a good ability to capture the spatial and temporal variations in SM and showing a correlation of approximately 0.60 with ISMN. SMOS IC and SMOS L3 followed in performance, while LPRM_C1, LPRM_C2, and LPRM_X exhibited relatively poor results in SM inversion. These findings serve as a valuable reference for improving satellite/reanalysis SM products and conducting global-scale SM studies.

**Keywords:** microwave remote sensing; soil moisture; SMAP; SMOS; LPRM; ERA5-Land





## 1. Introduction

Soil moisture (SM) plays a crucial role in the energy and water cycles of the land surface, influencing the exchange of energy, water, and carbon between the land and atmosphere, and influencing the distribution of surface energy and water resources [1–3]. Accurate SM data are essential for various applications, including drought monitoring, climate modeling, crop yield estimation, flood prediction, heat wave forecasting, and water resource management [4,5]. Standard field techniques can currently measure SM with an accuracy of up to 0.025 m³/m³ [6]. However, point-scale ground observations are limited in their ability to provide accurate data over large areas due to the low density of measurement points and the substantial spatial variability of SM [7,8]. Moreover, they

cannot effectively capture the spatial variability and heterogeneity of SM on a broader scale required for applications in this field [9,10].

To address the research and application objectives, the spatial estimation of SM using remote sensing satellites and models has emerged as the primary alternative for large-scale SM measurement [11–13]. These remote sensing satellites or sensors include the Soil Moisture Active Passive (SMAP) launched by NASA in 2015, the Soil Moisture and Ocean Salinity (SMOS) launched by ESA in 2009, and the Advanced Microwave Scanning Radiometer 2 (AMSR2) launched by JAXA in 2012 [14,15]. They offer advantages such as temporal repeatability, all-weather monitoring, and minimal disturbance to surface roughness [16,17]. The accuracy of satellite SM products is influenced by several uncertain variables, including microwave sensor characteristics, soil roughness, vegetation water content, climate, land cover conditions, and the inversion method. Model-derived SM products, such as the European Centre for Medium-Range Weather Forecasts Land Surface Reanalysis (ERA5-Land) SM product, offer advantages such as long-term coverage and the ability to monitor SM at various soil depths. However, these model products are constrained by input parameter uncertainties, atmospheric forcing errors, and model structure limitations. Therefore, it is essential to evaluate the performance of SM products before their use, in order to determine their suitable geographical areas of application and optimize the retrieval process [18,19].

Numerous validation studies have been conducted in recent years to assess the retrieval of satellite-based SM, including sparse and dense networks, core validation sites, post-launch campaigns, intercomparisons of remote sensing products, and ground model simulations [20,21]. For instance, Kim et al. (2018) [22] employed triple collocation (TC) analysis to compare SMAP, Advanced Scatterometer (ASCAT), and AMSR2, and found that SMAP exhibited the best spatiotemporal consistency with the reference data from April 2015 to December 2016. Al-Yaari et al. (2019) [23] compared multiple products, such as SMAP L3 V4, ASCAT H111, SMOS L3 V300, SMOS IC (V105), SMOS L2 V650, and Climate Change Initiative (CCI) V04.2, in diverse ecological zones and climatic conditions, revealing that climate, land cover types, and monitoring networks influence the performance of remote sensing SM retrievals. Zhang et al. (2019) [24] validated SMAP-enhanced L3 radiometer SM products across different climatic and landscape conditions using the International SM Network (ISMN) as a reference. Their findings indicated that SMAP still has room for improvement in areas with low surface temperatures and dense vegetation globally. Wigneron et al. (2021) [25] provided an overview of the latest scientific dataset for SMOS IC retrieval based on SMOS L-band observations, combining it with an earlier version (SMOS IC V105) to characterize the performance of the SMOS IC V2 product. Zheng et al. (2022) [26] assessed the accuracy of 24 different SM datasets using field measurements from 34 sites, determining that the newly developed SMAP-IB product without supplementary data exhibited the best performance based on bias statistics.

As microwave remote sensing has a short research cycle, SM retrieval algorithms are continually evolving and being revised and improved with new calibrations or concepts. Consequently, rigorous validation studies have been conducted on ten SM products, including SMAP DCA, SMAP SCA-H, SMAP SCA-V, SMAP-IB, SMOS IC, SMOS L3, land parameter retrieval models (LPRM_C1, LPRM_C2, and LPRM_X), and ERA5-Land products, across various global climates and landscapes. The objective of this current study was to evaluate the performance of these ten SM products from 1 January 2015, to 31 July 2022, and assess the influence of six parameters: climate zone, land cover, soil type, SM, leaf area index (LAI), and surface temperature on SM. Forty-five ISMN monitoring networks were utilized as ground measurement validation data. Additionally, the dynamic and static characteristics of the SMAP, SMOS, LPRM, and ERA5-Land products were further evaluated, considering the intercorrelations among the six aforementioned parameters.

## 2. Materials and Methods

### 2.1. Soil Moisture Data

#### 2.1.1. SMAP

SMAP, a dedicated satellite launched by NASA on 31 January 2015, is equipped with a synthetic aperture radar and an L-band radiometer to acquire global surface SM and freeze–thaw state data [27,28]. It operates in a near-polar sun-synchronous orbit, with ascending/descending passes at 18:00/06:00, and provides active, passive, and active-passive remote sensing SM products. These products have a 1000 km swath width, a fixed incidence angle of 40°, a revisit time of 2–3 days, and an accuracy of less than 0.04 cm$^3$/cm$^3$. However, due to a technical malfunction of the radar on 7 July 2015, the active and active-passive SM products had a shorter available time. Currently, SMAP products include an enhanced L3 radiometer dataset with a resolution of 9 km and a passive microwave observation dataset with a resolution of 36 km [29,30].

To enhance the accuracy of SM retrieval, the SMAP team has developed multiple retrieval algorithms, such as the V-pol single-channel algorithm (SMAP SCA-V), H-pol single-channel algorithm (SMAP SCA-H), dual-channel algorithm (SMAP DCA), and extended SMAP DCA. Additionally, SMAP-IB, developed by INRAE BORDEAUX, is a retrieval algorithm that utilizes dual-channel SMAP radiation observations without any modeling of SM data or input from optical vegetation indices [31].

In this study, we analyzed the applicability of SM products with a spatial resolution of 36 km, expressed in units of m$^3$/m$^3$, and projected them onto the EASE-Grid 2.0. This included daily SMAP L3 (SPL3SMP, version 8) products of SMAP SCA-V, SMAP SCA-H, and SMAP DCA, as well as the SMAP-IB (version 1) product. The SMAP L3 scientific data can be freely obtained in HDF format from the National Snow and Ice Data Center (NSIDC) (https://nsidc.org/data/smap/data; accessed on 22 July 2022), while the SMAP-IB data can be downloaded for free in NetCDF format from the INRAE BORDEAUX remote sensing product data center (https://ib.remote-sensing.inrae.fr/; accessed on 22 July 2022).

#### 2.1.2. SMOS

SMOS, a satellite launched by the European Space Agency on 2 November 2009, utilizes L-bands to measure global soil moisture. It employs multiple incidence angles, allowing observations at various angles ranging from 0° to 55°. SMOS monitors soil moisture near the surface at a depth of 0–5 cm, achieving an accuracy of 0.04 m$^3$/m$^3$ [32,33]. The satellite follows ascending and descending orbits at 06:00 and 18:00, respectively, with a temporal resolution of 2–3 days [34,35]. The global SMOS dataset currently consists of three grid products: SMOS Level 3 CADTS (SMOS L3), SMOS Level 3 BEC (SMOS-BEC), and SMOS Level 3 INRA-CESBIO (SMOS IC) [36]. In this study, we analyzed the SMOS L3 and SMOS IC soil moisture products with a spatial resolution of 25 km, projected onto the EASE-Grid 2.0. The L-band Microwave Emission of Biosphere (L-MEB) model was utilized for the analysis. SMOS scientific data can be obtained free of charge from the French CATDS Center (https://www.catds.fr/sipad/; accessed on 23 July 2022), and the data are provided in NetCDF format.

#### 2.1.3. LPRM

LPRM AMSR-2 is a land surface parameter retrieval model developed by the Vrije University Amsterdam, Netherlands, in collaboration with NASA. It is specifically designed for the estimation of soil moisture (SM) using passive microwave remote sensing data from AMSR-2 [37–39]. LPRM AMSR-2 utilizes both C- and X-band frequencies, and for this study, we selected the C1, C2, and X-bands with frequencies of 6.9 GHz, 7.3 GHz, and 10.7 GHz, respectively. The temporal resolution of the signal retrieval was set at 24 h, and the spatial resolution was 0.25°. The SM products employed in this research were LPRM_C1, LPRM_C2, and LPRM_X, with data version V001 [40–42]. The dataset containing these LPRM AMSR-2 SM products can be obtained free of charge from the Goddard Earth

Sciences Data and Information Services Center (GES DISC) (https://gcmd.gsfc.nasa.gov/; accessed on 23 July 2022), and it is provided in NetCDF format.

### 2.1.4. ERA5-Land

The ERA5-Land reanalysis SM data utilized in this study was derived from the fifth-generation reanalysis land surface dataset. It was obtained by downscaled ERA5 reanalysis data-driven ECMWF land surface model TESSEL, and it was made available by the European Centre for Medium-Range Weather Forecasts (ECMWF) [43,44]. This dataset provides SM with a temporal resolution of 1 h and a horizontal resolution of 0.1°, measured in volumetric water content ($m^3/m^3$). The SM data are divided into four layers based on depth: 0–7 cm, 7–28 cm, 28–100 cm, and 100–289 cm [45,46]. For this study, SM data from the 0–7 cm soil layer at 12:00 UTC were selected for analysis. The ERA5-Land reanalysis SM data can be obtained at no cost from the Copernicus Climate Change Service (https://cds.climate.copernicus.eu/cdsapp#!/home; accessed on 24 July 2022) in NetCDF format.

The start and end times of each remote sensing/reanalysis SM product data are presented in Table 1.

**Table 1.** Start and end dates for each remote sensing/reanalysis SM product.

| SM Products | Start Time | End Time |
| --- | --- | --- |
| SMAP DCA | 1 April 2015 | 10 July 2022 |
| SMAP SCA-H | 1 April 2015 | 10 July 2022 |
| SMAP SCA-V | 1 April 2015 | 10 July 2022 |
| SMAP-IB | 31 March 2015 | 31 March 2021 |
| SMOS IC | 31 March 2015 | 9 March 2022 |
| SMOS L3 | 31 March 2015 | 24 July 2022 |
| LPRM_C1 | 1 January 2015 | 31 July 2022 |
| LPRM_C2 | 1 January 2015 | 31 July 2022 |
| LPRM_X | 1 January 2015 | 31 July 2022 |
| ERA5-Land | 31 March 2015 | 30 April 2022 |

### 2.1.5. ISMN

To thoroughly assess the quality of SMAP, SMOS, LPRM, and ERA5-Land products, ground-based soil moisture (SM) measurements were obtained from the International Soil Moisture Network (ISMN), maintained by the Vienna University of Technology, Austria (https://ismn.earth/en/; accessed on 25 July 2022). Established in 2009, ISMN collects and manages in situ SM data globally from 1952 to the present, encompassing various depths and time ranges. Through its network infrastructure, ISMN automatically converts input field SM data into volumetric water content ($m^3/m^3$) while ensuring data quality control. ISMN offers free access to globally standardized in situ SM observations through its network interface, making it a centralized data system [47]. Presently, ISMN houses an extensive collection of in situ SM measurement datasets from approximately 72 networks and 2900 meteorological stations worldwide. Moreover, ISMN also stores auxiliary data, such as precipitation, climate type, land surface temperature, and land cover type, for select stations. These datasets have been widely utilized for the calibration, validation, and enhancement of remote sensing and reanalysis of SM products [48]. For this study, daily mean SM data from the top layer were selected from 45 networks and 2045 stations, covering the time range from 1 January 2015 to 31 July 2022. The spatial distribution of the sites is depicted in Figure 1.

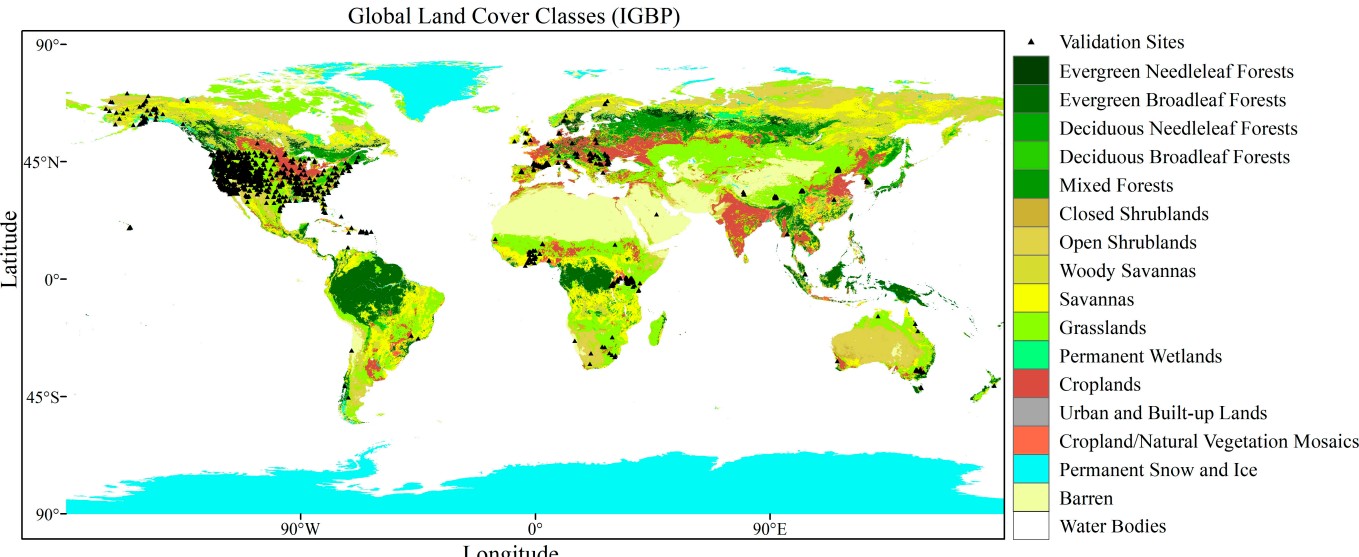

**Figure 1.** International Geosphere Biosphere Programme (IGBP) land cover classification and locations of different in situ SM measurements.

### 2.2. Static Conditions

In this study, the evaluation of the quality of SMAP, SMOS, LPRM, and ERA5-Land SM products was conducted under static conditions. Static conditions pertain to conditions that change relatively slowly and maintain a certain level of stability throughout the study period. The comprehensive assessment utilized ISMN data to evaluate the performance of these SM products across different climate zones, soil types, and land covers under static conditions.

#### 2.2.1. Climate Zone

This study utilized the Global Historical Climatology Network (GHCN) version 2.0 dataset to assess the suitability of each grid dataset across different climate zones. The GHCN dataset includes an updated map of the Koppen-Geiger climate classification, which offers a comprehensive representation of global climate zones [49]. This classification system categorizes the world's climate into five major types: tropical, arid, temperate, cold temperate, and polar. Each major type is further subdivided into 30 climate subtypes, based on local temperature and precipitation patterns. For this analysis, we focused on the primary classification of these climate zones.

#### 2.2.2. Land Cover Type

This study employed the International Geosphere Biosphere Programme (IGBP) classification map of land cover types, derived from the Moderate Resolution Imaging Spectroradiometer (MODIS) with a resolution of 10 km [48]. After excluding permanent wetlands, permanent ice and snow, and water areas, a total of 14 land cover types were retained. Among these, three land cover types (deciduous broadleaf forest, mixed forest, and closed shrubland) were selected for analysis as they had complete remote sensing/reanalysis SM product data. The aim was to assess the influence of different land cover types on the performance of SMAP, SMOS, LPRM, and ERA5-Land products.

#### 2.2.3. Soil Type

This study utilized the Global Soil Dataset, which serves as a comprehensive raster database integrating national and regional soil information. The database encompasses over 15,000 soil mapping units linked to soil property data, providing extensive coverage across the globe. It includes 26 distinct soil types, such as ANTHROSOLS, FLUVISOLS, KASTANOZEMS, GLEYSOLS, and CAL-CISOLS. For this study, sixteen soil types were

selected based on the availability of complete SM data and a wide range of products within each soil type. The selected soil types were LEPTOSOLS, VERTISOLS, FLUVISOLS, CAMBISOLS, LUVISOLS, CHERNOZEMS, PHAEOZEMS, PODZOLS, GREYZEMS, AN-DOSOLS, SOLONETZ, KASTANOZEMS, GLEYSOLS, ARENOSOLS, REGOSOLS, and PLANOSOLS. These soil types were utilized to evaluate the performance of SMAP, SMOS, LPRM, and ERA5-Land products across different soil types.

### 2.3. Dynamic Conditions

To assess the effects of dynamic factors on the retrieval of SM through remote sensing and reanalysis, this study examined the influence of variables that exhibit temporal variability within diverse climate zones. Specifically, the impact of measured SM, LAI, and LST on SM retrieval was analyzed. These dynamic factors, which vary over time, were considered within different climate zones to evaluate their influence on the accuracy of SM estimation.

#### 2.3.1. Soil Moisture

To assess the performance of satellite SM products, this study utilized ground-based SM observations that corresponded to the acquisition times of SMAP, SMOS, LPRM, and ERA5-Land. By evaluating the satellite SM products within different time intervals, the effectiveness and accuracy of these products could be assessed.

#### 2.3.2. Leaf Area Index

This study utilized the Modern-Era Retrospective Analysis for Research and Applications, version 2 (MERRA-2), LAI dataset, which has a temporal resolution of 1 month and a spatial resolution of $0.5° \times 0.625°$. MERRA-2 is the latest version of the satellite-era global atmospheric reanalysis produced by the Global Modeling and Assimilation Office (GMAO) of NASA, using the Goddard Earth Observing System Model (GEOS) version 5.12.4 [25]. The dataset covers the time range from 1980 to the present. It can be accessed for free in NetCDF format through the Goddard Earth Sciences Data and Information Services Center (GES DISC) at https://gcmd.gsfc.nasa.gov/; accessed on 26 July 2022.

#### 2.3.3. Land Surface Temperature

LST datasets corresponding to the depth of surface SM measurements in the ISMN were chosen to assess the suitability of various satellite SM products and reanalysis SM products across different LST conditions.

### 2.4. Other Auxiliary Data
NDVI

To understand the relationship between L-VOD products and vegetation. The Normalized Difference Vegetation Index (NDVI) dataset with a temporal resolution of one month and a spatial resolution of $0.05°$, specifically version MOD13C2 V061, was utilized in this study [25]. The dataset, obtained from NASA, covered the period from 2000 to the present. For this study, the time range selected was from January 2015 to July 2022. The dataset can be accessed free of charge through the Goddard Earth Science Data and Information Service Center (GES DISC) (https://gcmd.gsfc.nasa.gov/; accessed on 26 July 2022), and the data is provided in HDF format.

### 2.5. Data Processing

To assess the accuracy of satellite SM products and reanalysis SM products more effectively, data sites with relatively complete time series during the study period were chosen. The hourly surface SM and LST data from the ISMN were aggregated into daily averages. Each grid within the study area was treated as a station, and the nearest neighbor method was employed to extract the corresponding grid values from 10 datasets— SMAP DCA, SMAP SCA-H, SMAP SCA-V, SMAP-IB, SMOS IC, SMOS L3, LPRM_C1, LPRM_C2,

LPRM_X, and ERA5-Land—based on dynamic conditions (LST, LAI) and static conditions (climate zone, soil type, land cover type). For satellite data, the average value was used when there were no missing measurements for both ascending and descending orbits on the same day. If one orbit had missing measurements, the other orbit's value was considered valid. If both orbits lacked measurements, it was recorded as an invalid value. The remote sensing/reanalysis SM product was aggregated into daily, monthly, and annual average values. Finally, the SM data from the grid were compared with the observed SM data from the corresponding source monitoring station to verify the accuracy of the 10 gridded products.

Before conducting the verification, the data quality was controlled based on the quality control flags of the products. Firstly, the retained vegetation optical depth (VOD) values were restricted to the normal range of [0, 2], while for SM, the range was [0, 1]. Secondly, grids with freezing conditions (LST < 273 K) were excluded from SMAP DCA, SMAP SCA-H, SMAP SCA-V, SMAP-IB, SMOS IC, and LPRM datasets. Lastly, regions with vegetation water content (VWC) > 5 kg/m$^2$ were removed from SMAP DCA, SMAP SCA-H, and SMAP SCA-V.

### 2.6. Evaluation Metrics

#### 2.6.1. Coefficient of variation of the data series

To characterize the degree of variability in the SM changes, the coefficient of variation (CV) was selected [28]:

$$CV = \left| \frac{SD}{\overline{x}} \right| \tag{1}$$

where $SD$ represents the standard deviation and $x$ represents the mean. To characterize the degree of variability in SM changes, the $CV$ was used. The larger the $CV$, the more significant the changes in SM. A $CV < 10\%$ indicates weak variability, whereas values between 10% and 100% indicate moderate variability. Further, the indicator exhibited a strong variability when the $CV$ was >100%.

#### 2.6.2. Trend of Data Series

Using trend analysis to characterize the variation trend of SM, namely:

$$Slope = \frac{n \sum_{i=1}^{n} (i \times SM_i) - \sum_{i=1}^{n} i \times \sum_{i=1}^{n} SM_i}{n \sum_{i=1}^{n} i^2 - \left( \sum_{i=1}^{n} i \right)^2} \tag{2}$$

In the equation, slope represents the slope of the pixel regression equation, $SM_i$ is the SM value of the $i$-th year, and $n$ represents the length of the time series. A slope value greater than 0 indicates an increasing trend in SM, a slope value of 0 indicates no change in SM, and a slope value less than 0 indicates a decreasing trend in SM.

#### 2.6.3. Accuracy evaluation

ISMN observational data were used to verify the accuracy and applicability of 10 SM products globally—SMAP DCA, SMAP SCA-H, SMAP SCA-V, SMAP-IB, SMOS IC, SMOS L3, LPRM_C1, LPRM_C2, LPRM_X, and ERA5-Land. Four statistical indicators were selected to evaluate the accuracy of the SM products: the Pearson correlation coefficient (R), root mean square error (RMSE), bias, and unbiased root mean square error (ubRMSE). The following are the calculation formulas [50–52]:

$$R = \frac{\sum_{n=1}^{N} \left( SM_n^{sat} - \overline{SM^{sat}} \right) \left( SM_n^{obs} - \overline{SM^{obs}} \right)}{\sqrt{\sum_{n=1}^{N} \left( SM_n^{sat} - \overline{SM^{sat}} \right)^2} \sqrt{\sum_{n=1}^{N} \left( SM_n^{obs} - \overline{SM^{obs}} \right)^2}} \tag{3}$$

$$RMSE = \sqrt{\frac{1}{N} \sum_{n=1}^{N} \left( M_n^{obs} - SM_n^{sat} \right)^2} \tag{4}$$

$$Bias = \frac{1}{N} \sum_{n=1}^{N} (M_n^{obs} - SM_n^{sat}) \tag{5}$$

$$ubRMSE = \sqrt{RMSE^2 - Bias^2} \tag{6}$$

"$SM_n^{sat}$" and "$SM_n^{obs}$" refers to the satellite soil moisture value and ISMN observed soil moisture value, respectively. $\overline{SM^{sat}}$ represents the mean value of the *SM* product, whereas $\overline{SM^{obs}}$ represents the mean value of the ISMN. Lastly, "*N*" represents the total number of data.

## 3. Results

### 3.1. Evaluation Results under Static Conditions

3.1.1. Climate Zone

Figure 2 presents bar charts illustrating the overall performance results of SMAP DCA, LPRM_C1, LPRM_C2, LPRM_X, and ERA5-Land products across different climate zones. Additionally, the other five products exhibited poor performance in all five climate zones, with correlation coefficient (R) values below 0.3.

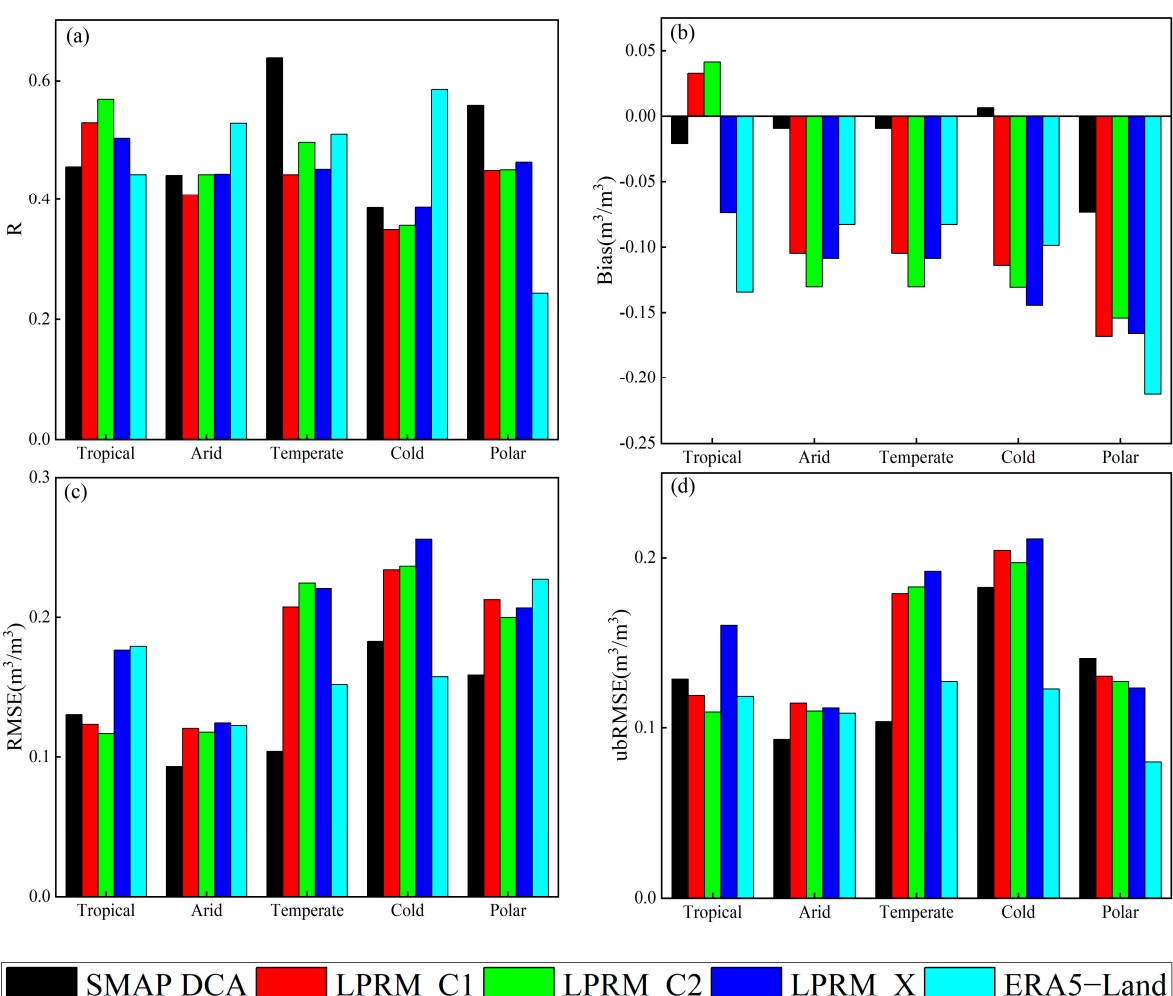

**Figure 2.** Remote sensing/reanalysis of SM products in different climate zones: (**a**) R, (**b**) Bias, (**c**) RMSE, and (**d**) ubRMSE.

Figure 2a displays the correlation coefficient (R) between the SM products and observed SM data. The LPRM series products demonstrated higher R values in the tropical zone, with LPRM_C2 achieving the highest correlation. In the arid zone, ERA5-Land

displayed the highest R value, while the other four products had similar R values. In the temperate zone, SMAP DCA exhibited the highest correlation (0.63) with the observed SM data, with comparable performance among the products. In the cold temperate zone, ERA5-Land exhibited the highest R value, whereas the other products showed significantly lower values compared to other climate zones. In the polar zone, the correlation between the ERA5-Land SM product and ISMN data decreased significantly, with SMAP DCA achieving the highest R value. In summary, LPRM_C2 performed the best in the tropical zone based on R values. Furthermore, ERA5-Land performed best in the arid and cold temperate zones and SMAP DCA in the temperate and polar zones.

Among the five climate zones, SMAP DCA exhibited the least absolute bias, approaching zero in the arid, temperate, and cold temperate regions, indicating superior performance. Positive biases were observed in LPRM_C1 and LPRM_C2 in the tropical region, as well as in SMAP DCA in the cold temperate region, suggesting underestimated SM values and dry soil conditions. On the contrary, the remaining products exhibited negative biases, indicating overestimation, which often increased with increasing latitude. Previous studies have shown that as vegetation density decreases [24], the brightness temperature monitored by the sensor also decreases, while the radiation signal has the opposite effect. This leads to significant differences between ground-based SM measurements and SM inversion in high-latitude regions.

Among the five climate zones, SMAP DCA exhibited the smallest root mean square error (RMSE), indicating its effectiveness in capturing spatial and temporal variability of surface SM at a global scale. Figure 2d presents the distribution of unnormalized RMSE (ubRMSE) values between remote sensing/reanalysis SM products and ISMN data. LPRM_C2 displayed the smallest ubRMSE in the tropics, SMAP DCA in the arid and temperate regions, and ERA5-Land in the cold temperate and polar regions. Based on ubRMSE, SMAP DCA and ERA5-Land outperformed the other products. Therefore, considering the various evaluation parameters mentioned above, SMAP DCA demonstrated better overall performance in most climate zones, with significant reliability in accurately capturing SM variations. Following SMAP DCA, ERA5-Land performed well, except in the Polar Regions. The performances of LPRM_C1, LPRM_C2, and LPRM_X were comparable, with minimal differences, and LPRM_C1 and LPRM_C2 slightly outperformed LPRM_X.

### 3.1.2. Land Cover Type

Figure 3 presents the validation metrics comparing ground-based SM observations and multisource remote sensing products for three land cover types. Figure 3a displays the correlation coefficient (R) values for closed shrubland, mixed forest, and deciduous broadleaf forests. Almost all SM products performed well in closed shrublands, with R values > 0.55. SMAP DCA, SMAP SCA-H, SMAP SCA-V, SMAP-IB, SMOS IC, LPRM_X, and ERA5-Land achieved R values > 0.7, which can be attributed to the lower and more uniform canopy in closed shrublands, making remote sensing monitoring of SM simpler. In deciduous broadleaf forests, SMAP series products and ERA5-Land exhibited higher R values (>0.45), with SMAP DCA and SMAP SCA-V showing similar R values (>0.62), enabling better detection of spatiotemporal SM variability. Notably, SMAP DCA performed exceptionally well in mixed forests, with an R value of 0.75, while the other products lagged, particularly the SMOS series and LPRM_C1 products, which struggled to capture SM changes effectively.

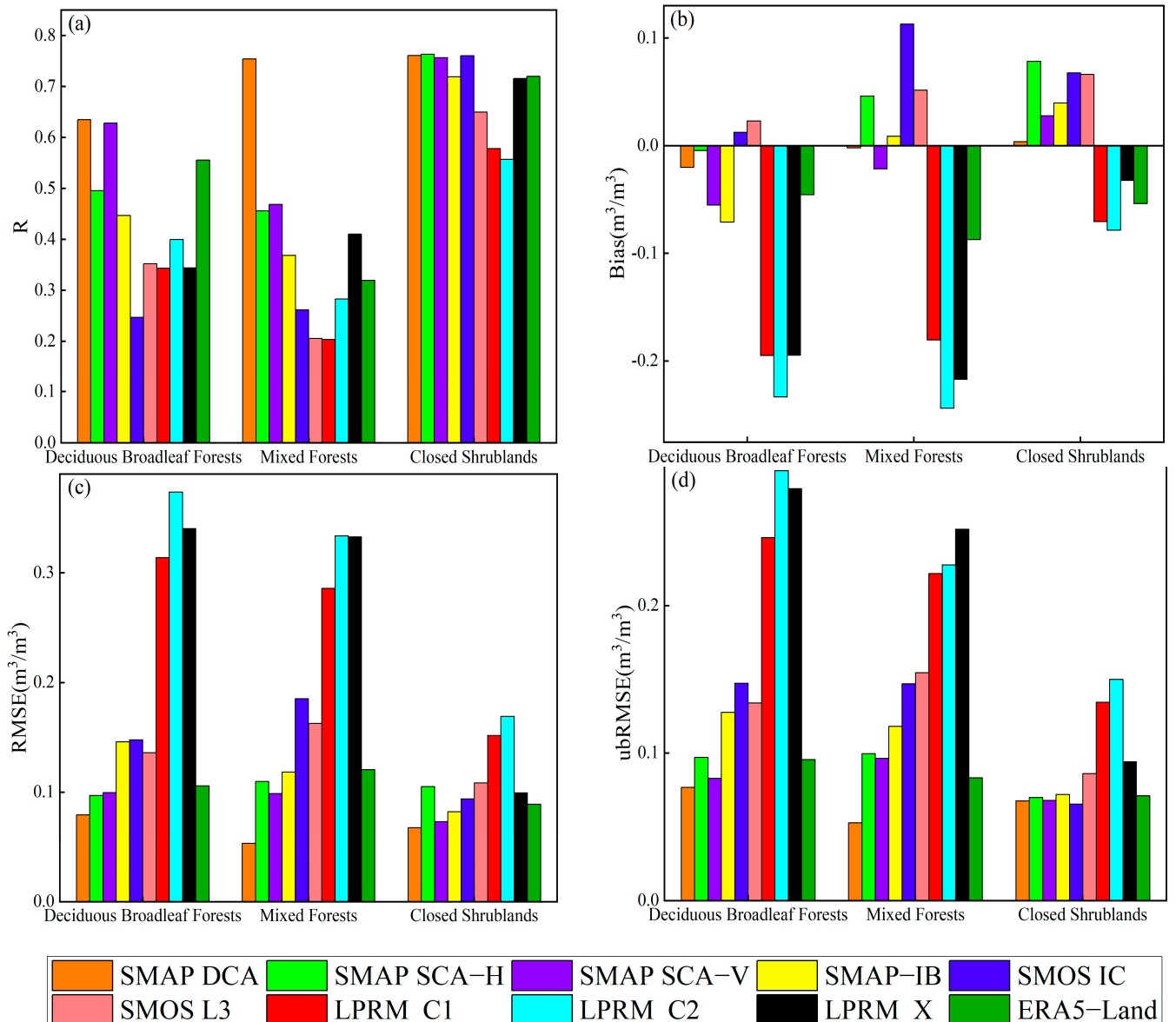

**Figure 3.** Performance of multisource SM products under different land cover types (Deciduous Broadleaf Forests, Mixed Forests, and Closed Shrublands). (**a**) R, (**b**) Bias, (**c**) RMSE, and (**d**) ubRMSE.

Figure 3b reveals significant bias differences between the same product and ground observations across different land cover types. In deciduous broadleaf forests, L-band SMAP DCA, SMAP SCA-H, SMAP SCA-V, and SMAP-IB exhibited varying degrees of negative bias (overestimation) compared to in situ values, whereas closed shrublands showed a positive bias (underestimation). The SMOS series products displayed a positive bias (underestimation) in all three land cover types. The LPRM_C1, LPRM_C2, LPRM_X, and ERA5-Land products exhibited negative biases in all three land cover types, resulting in an overestimation of SM.

Among the three land cover types, the L-band SMAP DCA product had the lowest bias, followed by the other SMAP products, SMOS products, and ERA5-Land. The performances of LPRM_C1, LPRM_C2, and LPRM_X in monitoring SM were comparable across the three land cover types, but they exhibited large absolute bias values, indicating wetter SM inversion. Compared to the C- and X-bands, the errors of the L-band satellite/reanalysis SM products in estimating SM for deciduous broadleaf forests, mixed forests, and closed shrublands were smaller, especially for the SMAP DCA product, which better captured

dynamic SM changes and demonstrated the best performance in monitoring global SM changes over time.

The distribution of root mean square error (RMSE) and unnormalized RMSE (ubRMSE) values were consistent for each grid dataset. Generally, RMSE and ubRMSE values were lower for closed shrub vegetation compared to other areas (Figure 2c,d), with each product performing better in this vegetation coverage area.

Considering all evaluation indicators, the SM products exhibited varying estimation accuracies, with the highest accuracy observed for the SMAP series, followed by ERA5-Land, SMOS series, and LPRM series products. Particularly, the SMAP DCA product demonstrated the best performance.

### 3.1.3. Soil Type

In this study, we employed Taylor diagrams to analyze the spatiotemporal variability and correlation of remote sensing/reanalysis SM products across 16 different soil types. This comprehensive evaluation aimed to assess the degree of agreement between various remote sensing/reanalysis SM products and ground measurements in different regions worldwide, while considering the effect of spatial heterogeneity. The diagrams also illustrated the relationship between the correlation coefficient (R) and standard deviation (SD) of remote sensing SM products and measured SM values at all sites.

Figure 4 displays the x- and y-axes representing the standard deviation of the remote sensing/reanalysis SM data, with the dashed line indicating the observation point. Points closer to the observation point indicate a better agreement between satellite remote sensing/reanalysis SM data and ground observation values. The radial lines represent the correlation coefficient between the remote sensing/reanalysis SM product and the field observation data. Scatter dots within the figure represent satellite SM products, excluding those with excessively large standard deviations that are not shown within the display range.

The scatter plots in the figure reveal that among the various soil types, SMAP DCA, SMAP SCA-V, SMAP-IB, and ERA5-Land products exhibit scatter plots closer to the observation points. They are followed by SMAP SCA-H, SMOS IC, and LPRM_C1, while SMOS L3, LPRM_C2, and LPRM_X products are further away from the observation points, indicating poorer performance in retrieving SM under different soil types.

The correlation coefficients between remote sensing/reanalysis SM and observation values in different soil types typically range from 0.3 to 0.8. This indicates varying degrees of ability to capture SM in different soil types based on correlation coefficients. The SMAP series products, ERA5-Land products, and ground observations show relatively stable correlation coefficients and outperform the other five products. However, LPRM_C2 exhibits a higher correlation coefficient in sandy soil (ARENOSOLS), planosol (PLANOSOLS), and LPRM_X shows a higher correlation coefficient in gray soil (PODZOLS), volcanic ash soil (ANDOSOLS), and SMOS IC. On the other hand, LPRM_C1 demonstrates lower correlation coefficients in all soil types, indicating suboptimal performance in capturing SM temporal changes.

Overall, SMAP-IB, ERA5-Land, SMAP DCA, and SMAP SCA-V products demonstrate the best overall performance in SM inversion, exhibiting a correlation coefficient of approximately 0.55 and a large R value. SMAP SCA-H, SMOS L3, and SMOS IC follow with an R value of approximately 0.43, while LPRM_C1, LPRM_C2, and LPRM_X perform less effectively, showing poorer correlation (approximately 0.40) and smaller R values.

### 3.2. Assessment Results of Dynamic Factors under Climatic Conditions

In this section, we analyzed the retrieval accuracy of various SM products for various dynamic ranges in five major climatic zones: tropical, arid, warm temperate, cold temperate, and polar regions. We also evaluated the accuracy of SM products in the presence of dynamic factors, such as the LAI and surface temperature.

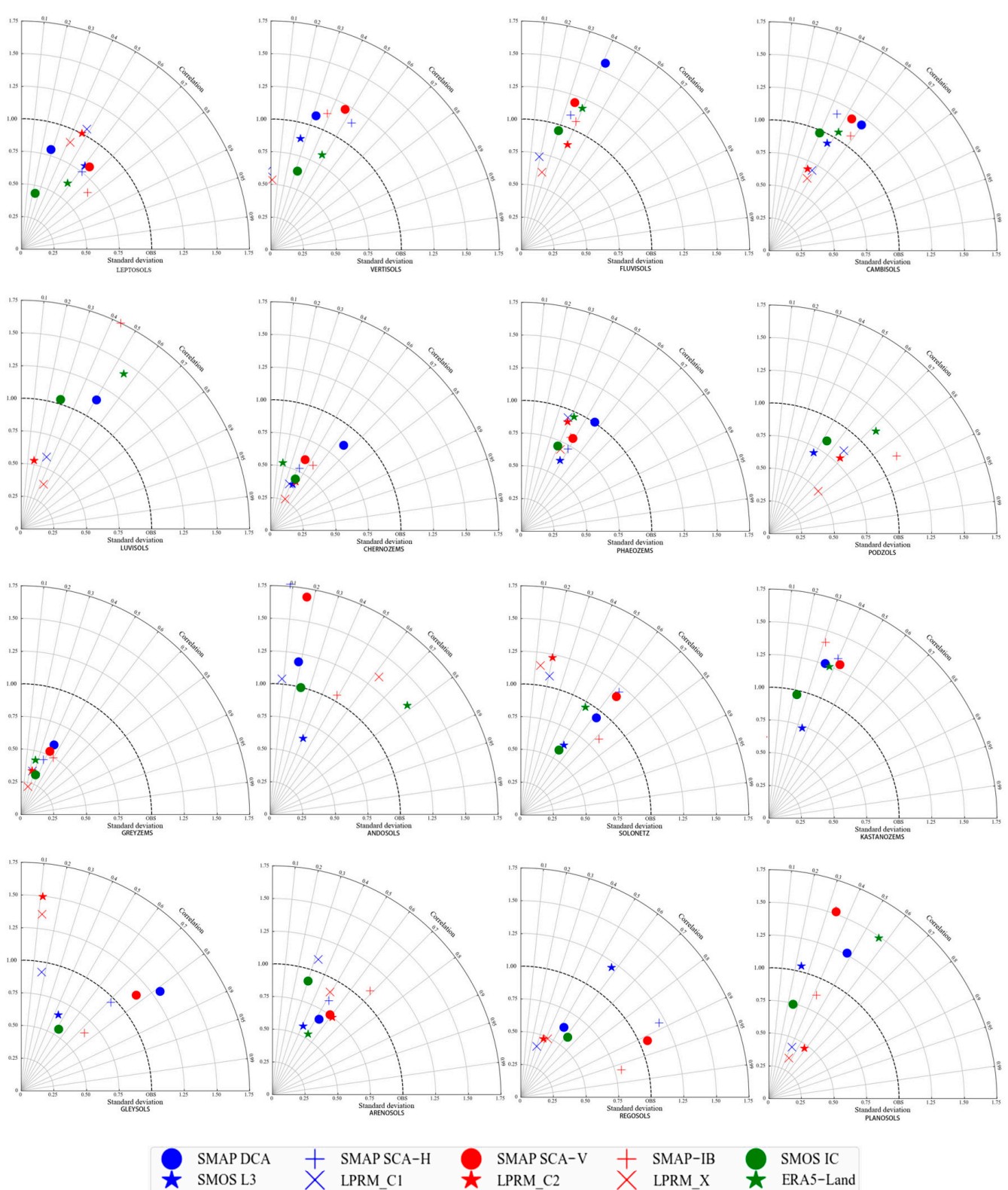

**Figure 4.** Taylor diagram showing the temporal performance of ground-based observed soil moisture and multisource satellite soil moisture product series across different soil types.

### 3.2.1. Soil Moisture

Figure 5 illustrates the retrieval accuracy of different SM products across various dynamic ranges in the five main climatic conditions. Generally, the correlation coefficients between satellite/reanalysis products and ISMN decrease as SM increases in tropical, arid,

and cold temperate regions. In warm temperate regions, the R value initially increases and then decreases with increasing SM. In polar regions, the R value increases as SM increases. Overall, compared to the other six products, SMAP-IB, SMAP DCA, SMAP SCA-H, and SMAP SCA-V exhibit relatively high correlation coefficients in various climate zones and are effective for monitoring global surface SM.

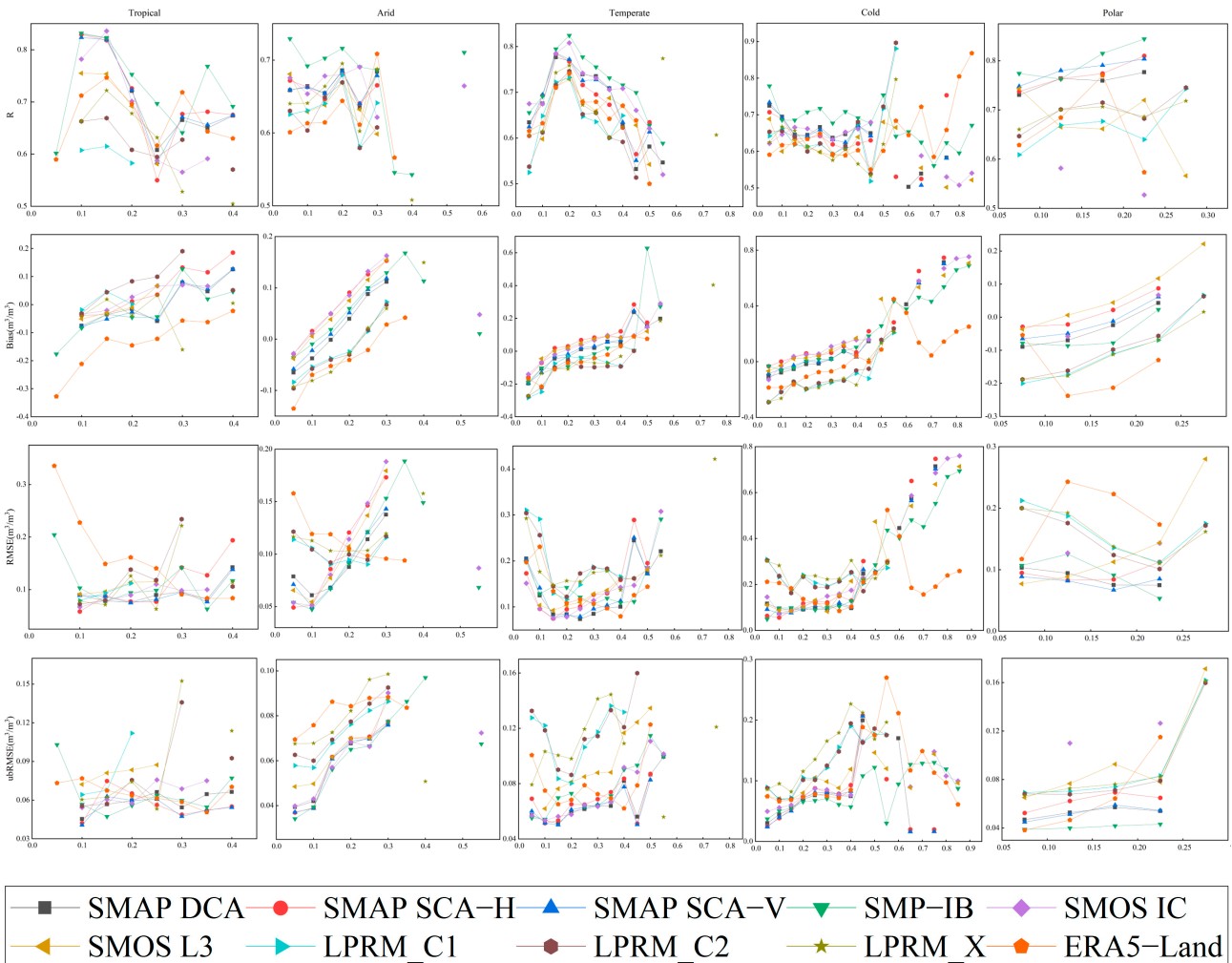

**Figure 5.** Retrieval performance of SM products under different climate conditions (horizontal axis represents SM in units of m$^3$/m$^3$).

In each climate zone, the SM products typically transition from negative to positive values as SM increases, indicating an increasing linear trend with a consistent rate of increase. This pattern suggests that remote sensing/reanalysis SM products tend to overestimate low soil moisture content while underestimating high soil moisture content. This finding aligns with a previous study [53].

The trend lines of SM products in each climate zone generally show a concave shape for the random error RMSE, with relatively minor variations in RMSE values ranging from 0.05 to 0.25 m$^3$/m$^3$, indicating stability. In tropical, warm temperate, and cold temperate zones, the inflection points with small RMSE values occur between 0.1 and 0.35 m$^3$/m$^3$, while in the arid zone and polar region, they fall between 0.1 and 0.2 m$^3$/m$^3$. Moreover, for SM values greater than 0.35 m$^3$/m$^3$ and 0.2 m$^3$/m$^3$ in the respective zones, the RMSE values increase significantly as SM increases.

The trend lines of the ubRMSE index for the SM products in each climate zone align with those of the RMSE. Among the SM products, SMAP-IB demonstrates the best

overall consistency with SM observations, followed by SMAP DCA, SMAP SCA-H, and SMAP SCA-V.

### 3.2.2. Leaf Area Index

Satellite/reanalysis SM products are significantly influenced by surface conditions, particularly vegetation characteristics, during SM retrieval. Therefore, calculating the distribution of statistical indicators of SM products in different LAI ranges allows for a comprehensive investigation of the influence of vegetation on SM. Figure 6 illustrates the distribution of evaluation indicators for different LAI ranges in each climatic zone based on satellite/reanalysis SM products and ISMN ground observation data. The figure reveals that LAI values in the warm and cold temperate zones exhibit a wide distribution range, typically ranging from 0 to 7.5. However, in the tropical and arid zones, the distribution range of LAI values is narrower, ranging from 0 to 3. The polar zone displays the narrowest distribution range of LAI values, ranging from 0.2 to 0.7.

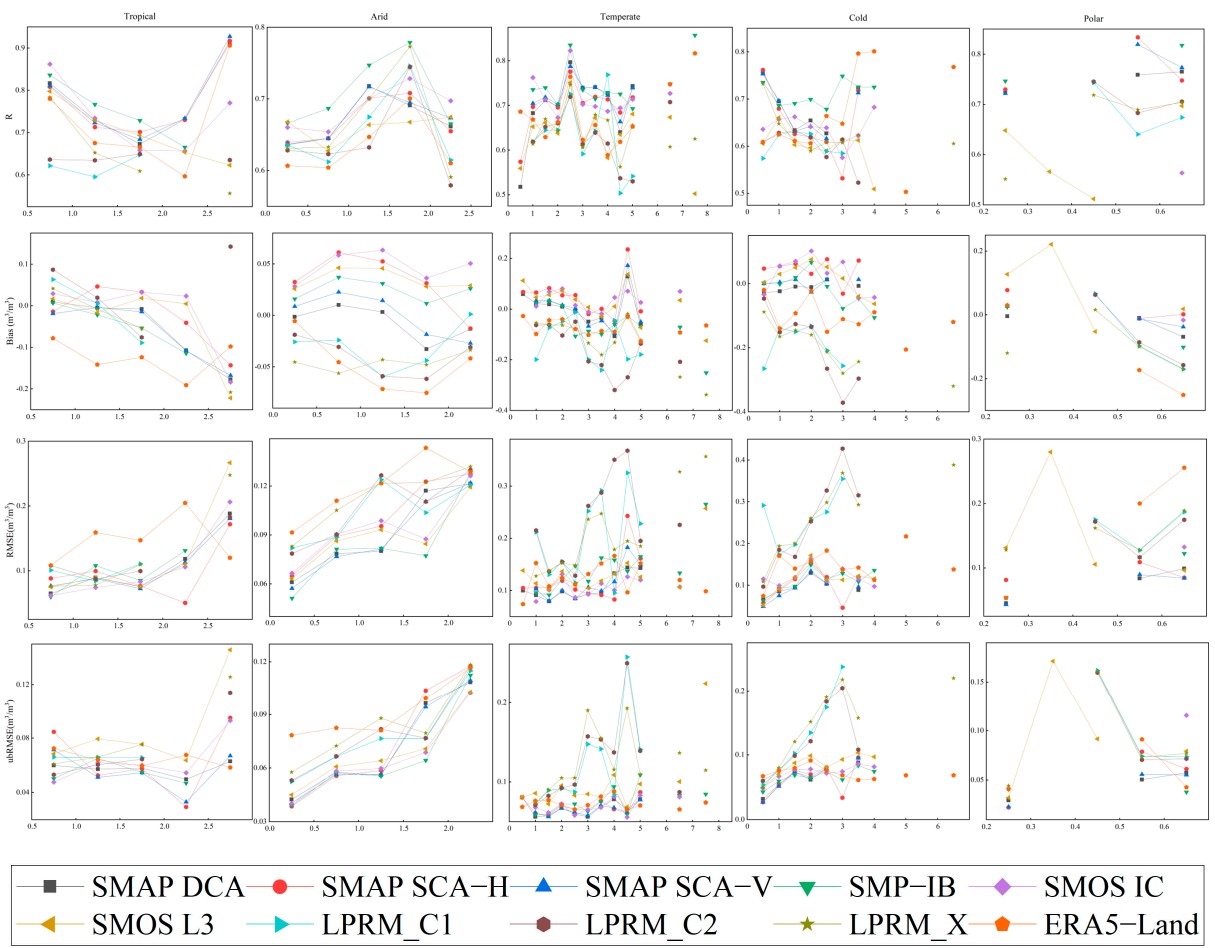

**Figure 6.** Performance of remote sensing/reanalysis products at different LAI values under climatic conditions (the horizontal axis represents LAI values in different intervals).

Correlation analysis of the data series shows a U-shaped curve for the correlation coefficient (R) between the SM products and ground observation data in the tropical zone, with a minimum valley at approximately 1.75. The R value is lowest when the LAI is around this value. Generally, the R value increases with a decrease or increase in LAI and can exceed 0.9. In the arid and temperate zones, the R value first increases and then decreases with rising LAI, remaining relatively stable when the LAI is between 1 and 3. In the polar zone, where vegetation coverage is minimal and site distribution is limited, the R value generally decreases with increasing LAI. SMAP-IB, SMAP DCA, SMAP SCA-H,

SMAP SCA-V, and SMOS IC exhibit high R values and perform well across all ranges of LAI compared to in situ SM.

Regarding bias indicators, most SM products exhibit a positive bias in tropical areas when the LAI value is between 0.5 and 1.0, indicating that the retrieved SM is drier than the actual value. As the LAI value increases, the bias gradually decreases from positive values (dry bias, underestimation) to negative values (wet bias, overestimation). In arid, warm temperate, and cold temperate regions, L-band SMAP SCA-H, SMAP-IB, and SMOS IC products generally show a positive bias, underestimating the SM. Conversely, C-band and X-band LPRM_C1, LPRM_C2, LPRM_X, and the ERA5-Land reanalysis product exhibit a negative bias, overestimating the SM. Moreover, when the LAI value is between 0.2 and 0.5, a positive bias is generally observed, while a negative bias is present when the LAI value is between 0.5 and 0.7.

The bias trends influence the RMSE and ubRMSE in different LAI intervals. As the LAI value increases, the RMSE and ubRMSE in the tropical, arid, warm temperate, and cold temperate zones generally exhibit an increasing trend, indicating that higher vegetation coverage and better water conditions lead to greater errors in SM inversion. The influence in the polar zone is insignificant due to limited ISMN observation data.

Overall, compared to other grid products, SMAP-IB, SMAP DCA, SMAP SCA-H, SMAP SCA-V, and SMOS IC exhibit good applicability under different vegetation cover conditions with minimal vegetation impact. For example, LPRM_X demonstrates reliability in dry areas with low vegetation cover, whereas it is more affected by vegetation in areas with higher vegetation cover.

### 3.2.3. Land Surface Temperature

The influence of land surface temperature (LST) on the brightness temperature signal on the ground should be considered in satellite/reanalysis SM retrieval techniques. Figure 7 illustrates the performance of SM products under different LST ranges in various climate zones to assess the impact of LST on product accuracy. Due to the lack of LST data in tropical, arid, and warm temperate zones, the analysis of the impact of LST on SM inversion was limited to cold temperate and polar conditions.

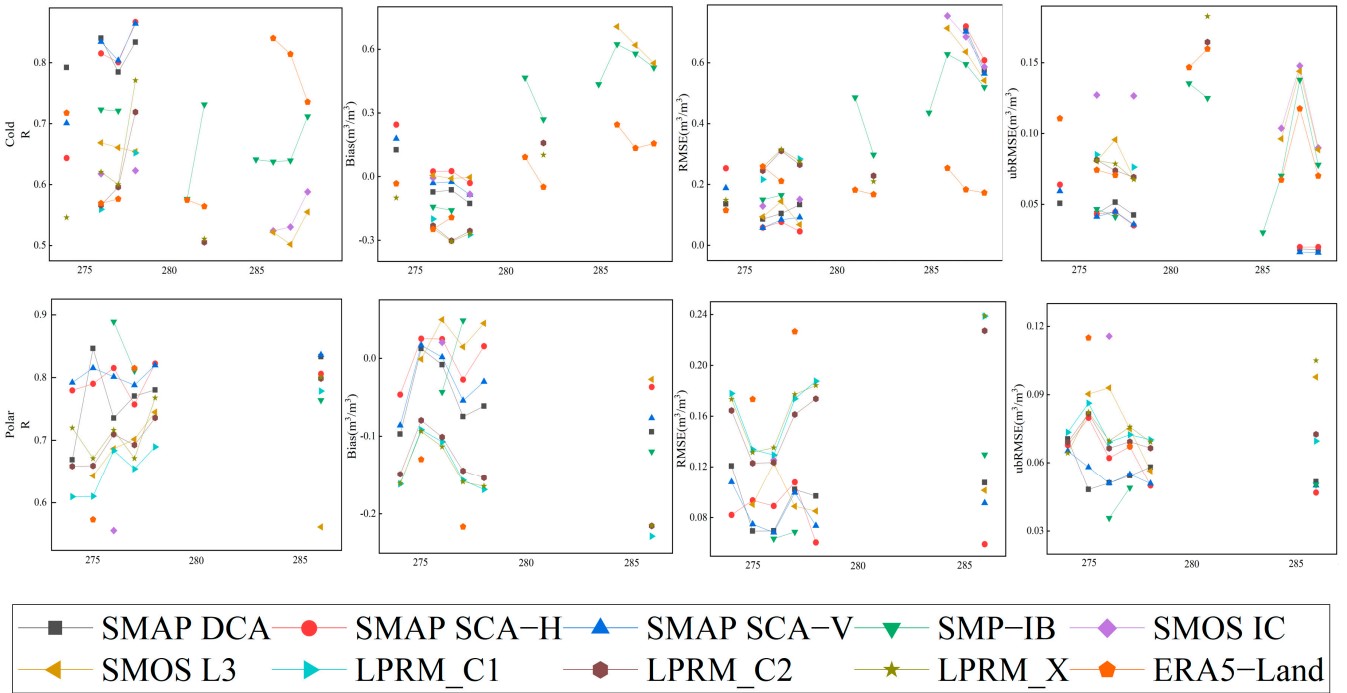

**Figure 7.** Performance of remote sensing/reanalysis products at different LST values (*x*-axis represents different intervals of LST values in K) under climatic conditions.

Based on the correlation analysis of the data series, the highest correlation coefficient between SMAP SCA-V and ISMN ground observation data was observed in the cold temperate zone. In contrast, the correlation coefficients between SMOS L3, LPRM_C2, and ground observation data were relatively small. In the polar zone, SMAP-IB exhibited the highest correlation coefficient, while the correlation coefficients of SMOS IC and LPRM_C2 data series were relatively small. In the cold temperate and polar zones, the predominant range of correlation between each SM product and ISMN data was 0.6 to 0.8, indicating good SM retrieval performance. The R values of the individual products increased with increasing LST, although the change was insignificant.

SMAP SCA-H, SMAP SCA-V, and SMOS L3 showed relatively small biases in cold and temperate regions, while LPRM_C1, LPRM_C2, and LPRM_X exhibited negative biases, indicating a bias toward wet conditions in the retrieved SM. The changes in RMSE and ubRMSE exhibited similar trends, with generally better values in polar regions compared to cold temperate regions.

In the range of land surface temperatures in cold temperate and polar regions, SMAP-IB, SMAP DCA, SMAP SCA-H, and SMAP SCA-V outperformed other products. This is attributed to the SMAP retrieval algorithm's utilization of LST correction to improve data accuracy. Additionally, as LST increased in cold temperate and polar areas, the impact of LST on the information content of SM in the L-band was relatively minimal, whereas it had a more significant impact on the C-band, X-band, and ERA5-Land.

*3.3. Global Spatial Patterns of Soil Moisture Changes*

Figure 8 illustrates the spatial distribution pattern of the multi-year average SM inversion of satellite/reanalysis SM products between 2015 and 2022. Based on the quality control standards of the products, areas with VWC > 5 kg/m$^2$ were excluded, resulting in a lack of data for SMAP DCA, SMAP SCA-H, and SMAP SCA-V products near the equator at low latitudes. Similarly, areas with LST < 273 K for SMAP series products, SMOS IC, and LPRM series products were excluded, leading to the absence of data in certain high-latitude areas. Unlike the reanalysis product ERA5-Land, other SM products did not provide data for the Antarctic region, which is represented as white or zero values.

At a global scale, the SMAP and SMOS products exhibited similar spatial patterns in SM distribution. Tropical rainforest areas, such as the Amazon Basin, Congo Basin, Malay Archipelago, and their surrounding regions, displayed the highest SM values ranging from 0.4 to 0.9 m$^3$/m$^3$. In contrast, tropical desert areas in arid and semi-arid regions such as the Sahara Desert, central and western Oceania, and Central Asia exhibited the lowest SM values (below 0.1 m$^3$/m$^3$). SM values were also comparable in low-latitude tropical grasslands, mid-latitude Mediterranean coastal areas, and high-latitude temperate continental climate zones.

The spatial distribution patterns of the LPRM_C1, LPRM_C1, and LPRM_X products were consistent. These products demonstrated an inverted arid-wet gradient between tropical rainforest areas, such as the Amazon and Congo basins, and the Appalachian region of the eastern United States. Theoretically, SM values in tropical rainforest areas should be higher than in the Appalachian region. However, the figure shows the opposite trend. Similar situations were observed in high-latitude regions of the Northern Hemisphere and the Qinghai–Tibet Plateau region of China, where SM values should theoretically be lower due to weaker evaporation and a greater distance from the ocean. Nevertheless, the figure depicts an opposite trend in the distribution of SM values. The maximum SM value was located in the high-latitude regions of the Northern Hemisphere, followed by the Qinghai–Tibet Plateau region and subtropical monsoon regions, with tropical rainforest regions having the next highest values and tropical desert regions exhibiting the lowest values.

The SM values of the ERA5-Land product were generally higher. However, excluding tropical desert areas, SM values decreased from the equator towards the north and south, with higher SM concentrations observed in the Arctic Ocean compared to Antarctica.

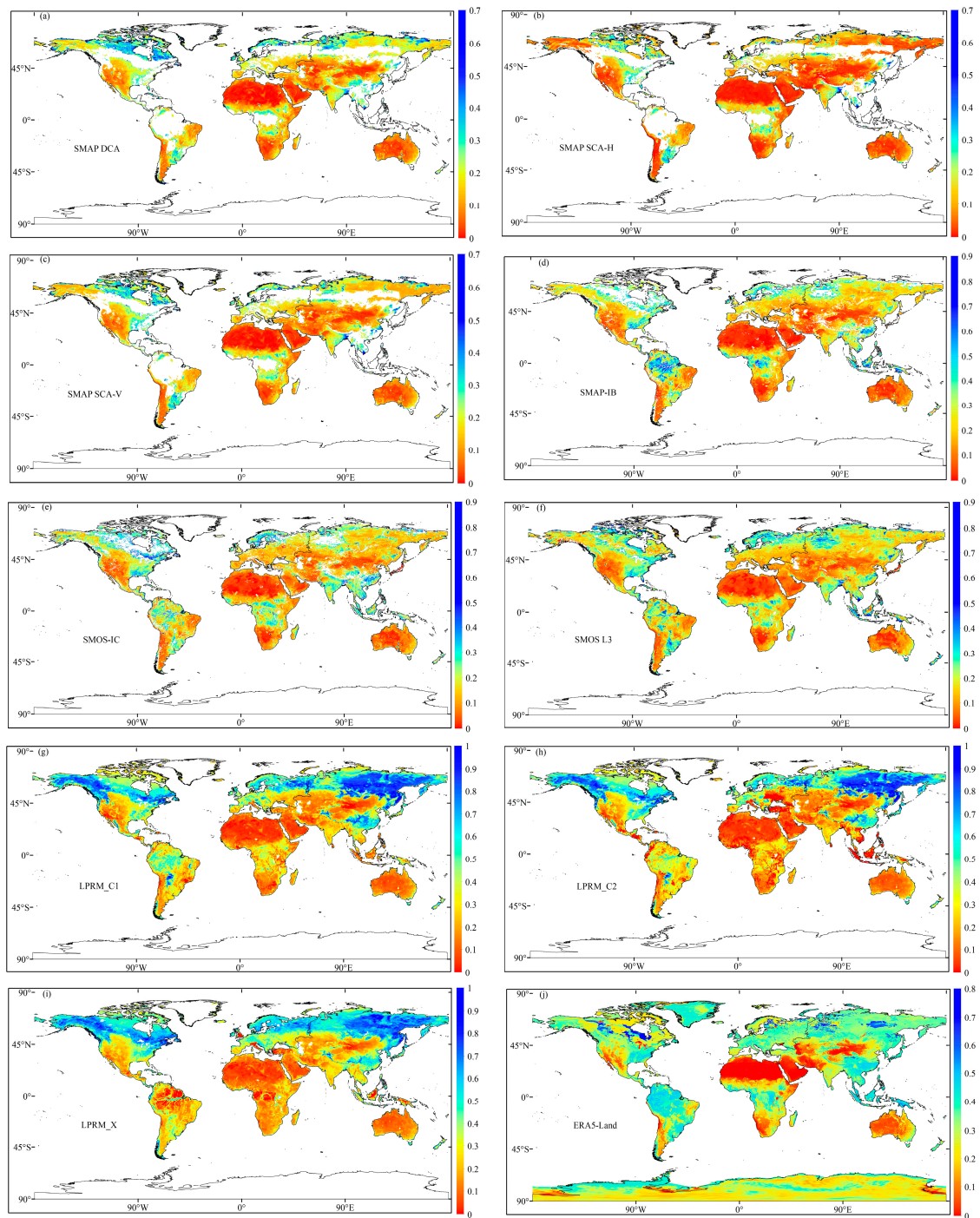

**Figure 8.** Spatial distribution of multi-year average SM remote sensing products and reanalysis product retrievals. (**a**) SMAP DCA, (**b**) SMAP SCA-H, (**c**) SMAP SCA-V, (**d**) SMAP-IB, (**e**) SMOS IC, (**f**) SMOS L3, (**g**) LPRM_C1, (**h**) LPRM_C2, (**i**) LPRM_X and (**j**) ERA5-Land.

In general, dry areas included the Sahara Desert, Arabian Peninsula, central and eastern Iran, northwestern China, the western United States, central and western Australia, and southwestern South America. These regions either represented subtropical desert areas controlled by subtropical high-pressure systems with high temperatures and low rainfall throughout the year or deep inland areas far from the ocean where water vapor is scarce. On average, the northern high-latitude regions, including Canada, Europe, and Russia, were also drier with the L-band SMAP and SMOS series products compared to the C- and X-band LPRM series products.

Figure 9 illustrates the spatial variation trend of SM between 2015 and 2022. Most areas showed an increasing trend in SM based on the data from the SMAP DCA, SMAP SCA-V, SMAP-IB, SMOS IC, and ERA5-Land products. The increase was particularly notable in the Great Plains regions of the central United States, the Mediterranean coast, the Middle East, North African desert areas, and grassland areas of southern Africa, Asia, eastern South America, and northern Oceania. The SM trends for these five products were as follows: 57.43%, 57.27%, 51.73%, 53.99%, and 63.48% increase, respectively. The ERA5-Land product exhibited the most pronounced trend. Significant contributions to the rising SM ($p < 0.05$) were observed from the eastern and western regions of South America and the southern and northern regions of Africa.

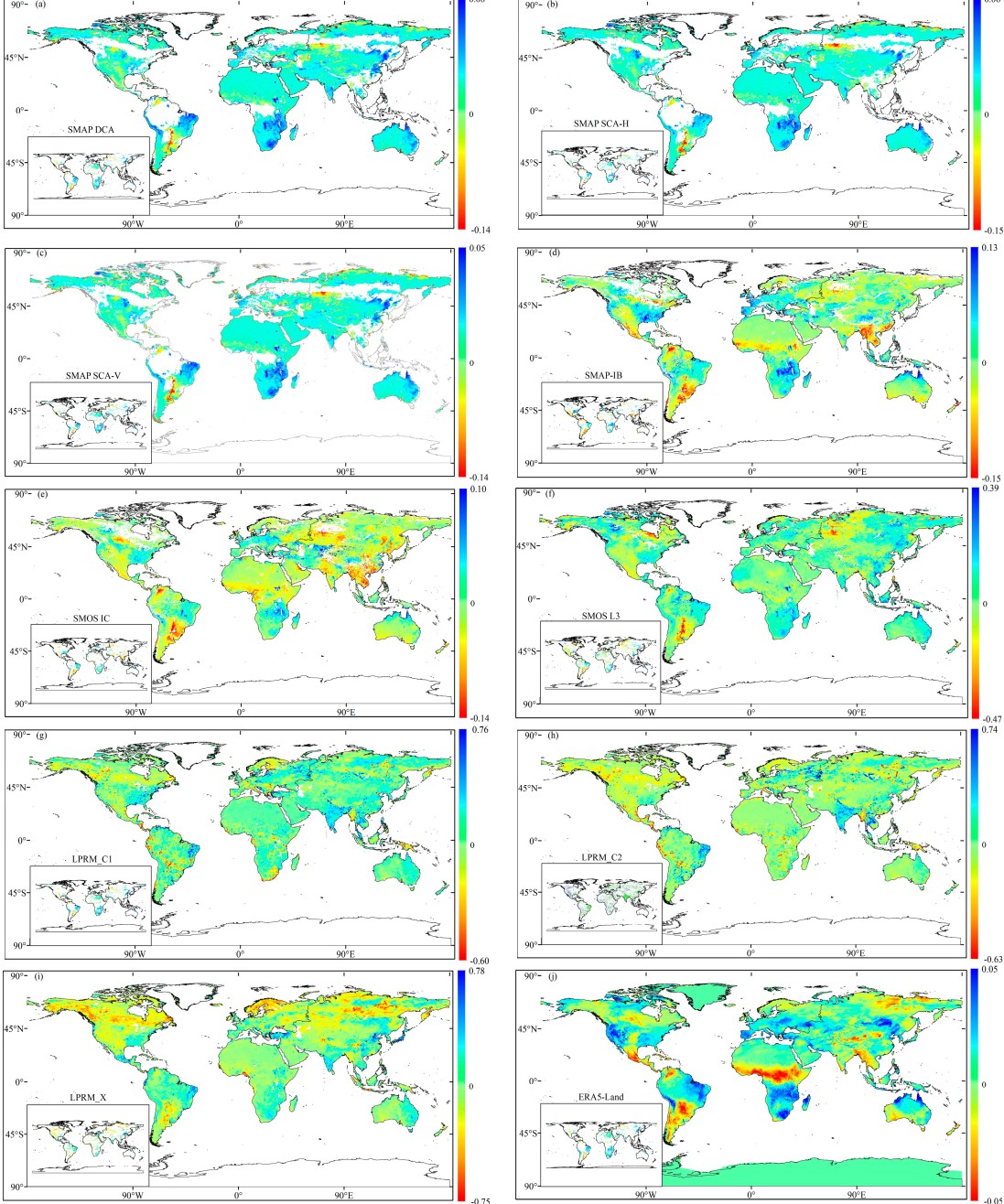

**Figure 9.** 2015–2022 annual trend in SM from multiple sources (m$^3$/m$^3$) (significance test in the lower left corner). (**a**) SMAP DCA, (**b**) SMAP SCA-H, (**c**) SMAP SCA-V, (**d**) SMAP-IB, (**e**) SMOS IC, (**f**) SMOS L3, (**g**) LPRM_C1, (**h**) LPRM_C2, (**i**) LPRM_X and (**j**) ERA5-Land.

In contrast, the SMAP SCA-H, SMOS L3, LPRM_C1, LPRM_C2, and LPRM_X products demonstrated a decreasing trend in SM, covering 50.76%, 54.89%, 54.58%, 53.45%, and 55.29% of the global area, respectively. These areas were mainly distributed in high-plateau regions of southern South America, the Rocky Mountains in the western United States, the Congo Basin in central Africa, inland areas of Asia far from the ocean, and desert areas in southwestern Oceania. Other regions with decreasing SM included the South African Plateau, the Indochinese Peninsula, and northern Asia. Notably, the changes in SM differed significantly between South Asia and the Indochina Peninsula, which are influenced by a tropical monsoonal climate. While SM retrieved from SMAP-IB, SMOS IC, and ERA5-Land exhibited a downward trend, the other seven products showed an opposite trend.

Overall, the SM trends observed from 2015 to 2022 using various data sources were generally consistent. An increasing trend was observed along Brazil, Madagascar, Kuroshio, and East Australian currents, while a decreasing trend was observed along the Benguela, California, and Peru currents.

Figure 10 presents the statistical results of the trend analysis for SM products. From 2015 to 2022, more than half of the global area exhibited increasing SM trends for products such as SMAP DCA, SMAP SCA-V, SMAP-IB, SMOS IC, and ERA5-Land, with significant increases of 14.43%, 15.04%, 10.30%, 10.61%, and 0.49% of the total area, respectively. The analysis of SM extraction from SMAP DCA, SMAP SCA-V, SMAP-IB, and SMOS IC revealed that 42.27% to 48.27% of the worldwide area showed a drying trend, while ERA5-Land indicated a drying trend in 36.52% of the areas.

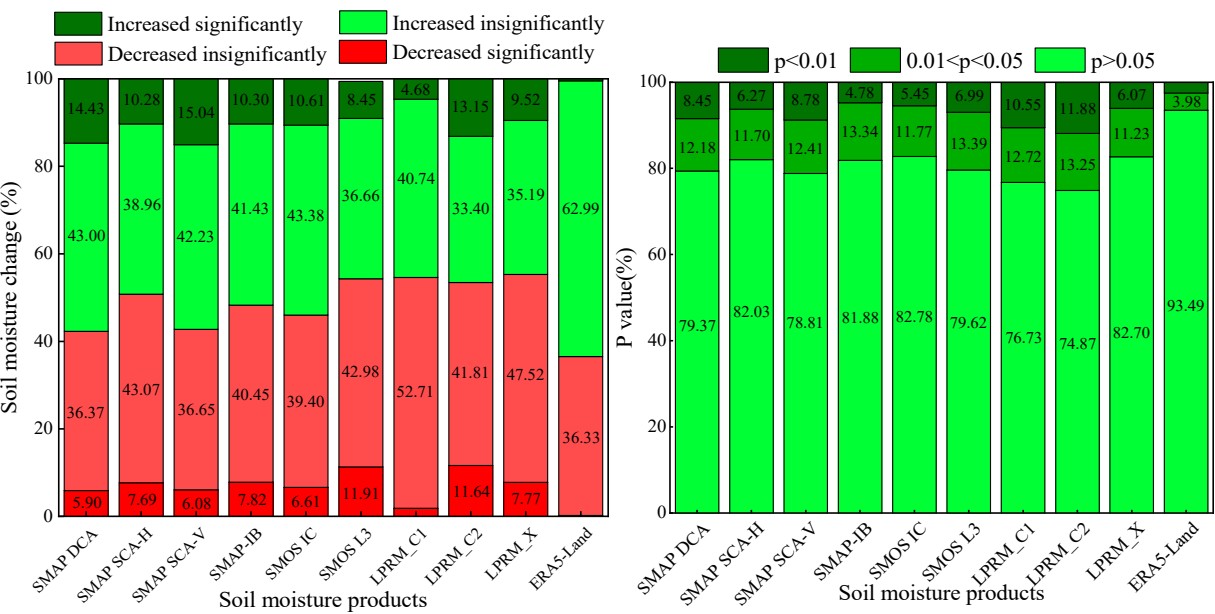

**Figure 10.** SM trend and percentage of *p*-values from 2015 to 2022.

In contrast, SMAP SCA-H, SMOS L3, LPRM_C1, LPRM_C2, and LPRM_X products demonstrated a declining trend in SM in more than half of the global area from 2015 to 2022. The areas with a significant decreasing trend accounted for 7.69%, 11.91%, 1.87%, 11.64%, and 7.77% of the total area, respectively. Analysis of the SM data extracted from these five products revealed that 50.76% to 55.29% of the global area exhibited a drying trend from 2015 to 2022.

In terms of statistical significance, the proportion of *p*-values for the numerical trend of each satellite/reanalysis SM product globally was lowest at *p* < 0.01 (extremely significant), followed by 0.01 < *p* < 0.05 (significant), and the proportion of *p* > 0.05 (insignificant) was highest.

Figure 11 depicts the spatial distribution of the coefficient of variance (CV) of soil moisture (SM) from 2015 to 2022. In the L-band, the CV of the SMAP and SMOS series

products ranged from 0 to 2.4, with most regions exhibiting CV values between 0.1 and 1. These values indicate moderate variability and generally stable SM content. Regions such as tropical grasslands in Africa, South America, and Oceania exhibited the highest CV values, while subtropical evergreen broadleaf forests were found along the Mediterranean coast, and temperate grasslands were observed in Asia. Low CV values were mainly observed in tropical rainforests in the Amazon Plain, Congo Basin, and the Malay Archipelago, as well as in tropical monsoon rainforests in Southeast Asia, evergreen broadleaf forests in eastern Asia and the eastern United States, and high-latitude regions.

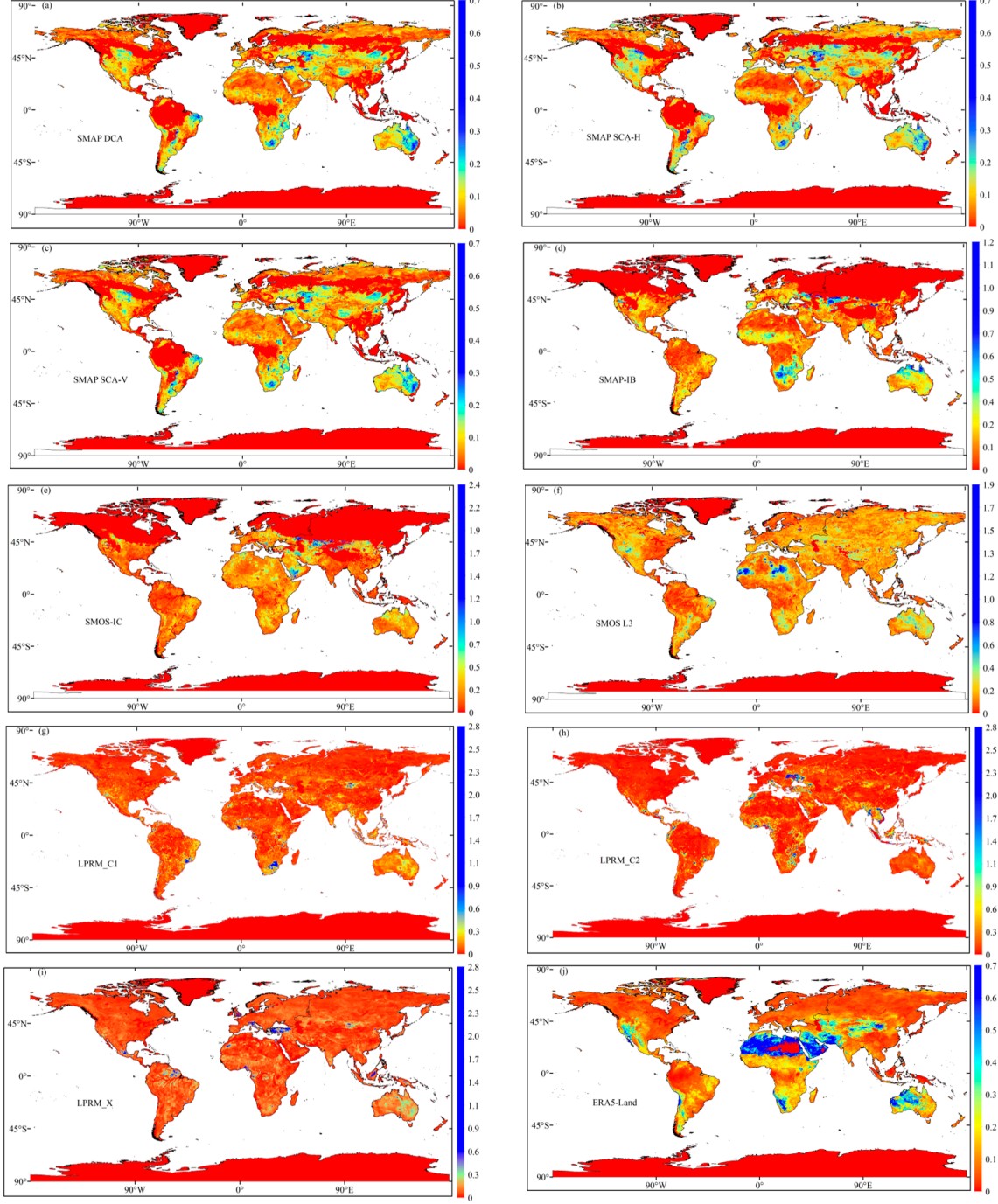

**Figure 11.** The spatial distribution of the coefficient of variation of SM data from various sources from 2015 to 2022. (**a**) SMAP DCA, (**b**) SMAP SCA-H, (**c**) SMAP SCA-V, (**d**) SMAP-IB, (**e**) SMOS IC, (**f**) SMOS L3, (**g**) LPRM_C1, (**h**) LPRM_C2, (**i**) LPRM_X and (**j**) ERA5-Land.

In the C- and X-bands, the maximum CV for SM in the LPRM_C1, LPRM_C2, and LPRM_X products was 2.8, and the spatial distribution of CV was comparable. High CV values (>1) were primarily found in the Mediterranean and tropical rainforest climates, indicating weak content stability and significant variability.

The regions with the highest coefficient of variance for ERA5-Land SM were the tropical desert climate region and the temperate continental climate region in central and western Asia, demonstrating a moderate degree of variation. Conversely, the tropical rainforest climate region, tropical monsoon climate region, and high-latitude regions exhibited low CV values, indicating weak variability. The SM changes in these regions were relatively stable.

## 4. Discussion

This study aimed to analyze and evaluate the applicability of 10 soil moisture (SM) products at a global scale. The products included SMAP DCA, SMAP SCA-H, SMAP SCA-V, SMAP-IB, SMOS IC, SMOS L3, LPRM_C1, LPRM_C2, LPRM_X, and ERA5-Land. The findings revealed that the L-band and reanalysis SM grid products outperformed the C/X-band SM products under both static (climate zone, land cover type, and soil type) and dynamic (in situ SM, LAI, and surface temperature) conditions. Additionally, they performed better than ISMN across multiple evaluation indicators and were able to more accurately capture actual SM values.

These findings are consistent with previous studies. Min et al. [54] found that SMAP and ESA CCI demonstrated the best overall performance under dynamic conditions (LST, SM, and VOD) at a global scale, followed by SMOS L3 and SMOS IC. AMSR2 and FY3B, on the other hand, performed poorly. Zhang [55] revealed that SMAP and SMOS IC exhibited consistently high temporal correlations and significant reproducibility of SM variability across climate zones, land cover types, and VWC conditions globally.

In SM retrieval algorithms, the vegetation effect has a significant impact on simulated brightness values obtained from land surface models and field measurements. Therefore, L-VOD (vegetation optical depth) can be an effective substitute for vegetation correction. Furthermore, the L-VOD retrieved by the SMAP algorithm is less affected by radio frequency interference (RFI), which can enhance the accuracy of remote sensing products and SM model simulations at a global scale. In this study, we calculated the high-frequency variation SD (SDHF) of L-VOD products and analyzed their correlation with NDVI, VWC, and LAI, providing insights into environmental changes.

Figure 12 illustrates the SDHF of four sets of L-VOD time series from SMAP DCA, SMAP SCA-H, SMAP SCA-V, and SMAP-IB. This quantified the variability of the L-VOD product between 2015 and 2022. The spatial distribution pattern of SDHF was similar for SMAP SCA-H and SMAP SCA-V. Globally, the SDHF of L-VOD products retrieved by the SMAP DCA algorithm was higher compared to the other three products. In tropical forest areas such as the Amazon and Congo Basins, SMAP-IB had higher SDHF values, while the other three products exhibited lower SDHF values.

The SDHF of these four L-VOD products was high in central North America, Europe, the Asian monsoon region, northern Asia, Sub-Saharan Africa, south of the Saharan Desert, the Brazilian plateau in South America, and the southern part of Oceania. Precipitation significantly influences vegetation growth in regions such as Mexico, the United States, eastern Brazil, southern Australia, India, northern Russia, and eastern and southern China, leading to relatively high SDHF values. These variations can introduce errors in the inversion of SM by remote sensing/reanalysis products caused by vegetation, which aligns with the findings of other studies [24]. It has also been discovered that in areas with large SDHF values in India and China, significant variances are attributed to the influence of RFI [25]. Additionally, variations in acquisition time and sensor calibration may contribute to variances in the SDHF of remote sensing data.

Figure 13 illustrates the spatial pattern of the temporal correlation between the L-VOD products obtained from the SMAP DCA, SMAP SCA-H, SMAP SCA-V, and SMAP-IB

algorithms and NDVI. Overall, the global correlations between the four L-VOD products and NDVI were consistent, indicating changes in vegetation growth and greenness across space. Higher correlations were observed in the mid-latitude regions of the Northern Hemisphere, including the Midwestern United States, Europe, and the monsoon regions of Asia. Significant correlations were also found in the Southern Hemisphere, specifically between the East African Plateau, South African Plateau, Brazilian Plateau, and the northern and southern Australian plains. The correlations between SMAP DCA, SMAP SCA-H, SMAP SCA-V L-VOD, and NDVI could exceed 0.9. The correlation between SMAP SCA-H and SMAP SCA-V with NDVI was stronger compared to that of SMAP DCA. This difference arises because the SMAP DCA algorithm does not utilize MODIS NDVI climatic data as a reference for the time series and is unrelated to optical vegetation indices. On the other hand, the SMAP SCA-H and SMAP SCA-V algorithms require reference to MODIS NDVI data when estimating L-VOD, resulting in a higher correlation coefficient with NDVI [53].

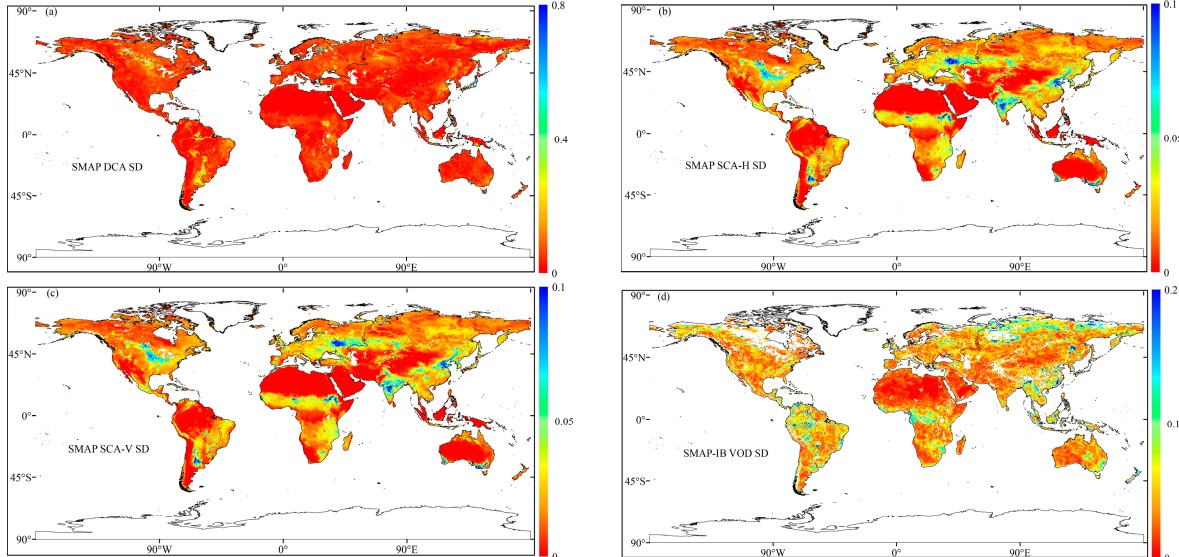

**Figure 12.** Standard deviation maps of global high-frequency variations for different L-VOD products. (**a**) SMAP DCA VOD, (**b**) SMAP SCA-H VOD, (**c**) SMAP SCA-V VOD and (**d**) SMAP-IB VOD.

In tropical rainforests, tropical deserts, and high-latitude locations with low interannual vegetation dynamics, the spatial and temporal correlations among SMAP DCA, SMAP SCA-H, SMAP SCA-V, SMAP-IB, and NDVI were low. Previous studies have revealed that in many forest ecosystems, the low spatiotemporal correlation between L-VOD and NDVI is attributed to the asynchrony of plant water storage and leaf development. Consequently, the quality of L-VOD cannot be accurately determined using woody vegetation types. Therefore, when calculating the correlation between L-VOD and NDVI in low-vegetation ecosystems, it is essential to consider that the two indices exhibit different sensitivities to various vegetation characteristics [25].

Figure 14 displays density plots depicting the correlations between the SMAP DCA, SMAP SCA-V, and SMAP-IB L-VOD products and VWC, LAI, and NDVI. Since the SMAP SCA-H and SMAP SCA-V L-VOD products exhibit high similarity, only one of them is discussed. The figure reveals a significant linear relationship between L-VOD and VWC, as well as between L-VOD and LAI, with high correlation coefficients. The spatial correlation (R values) between L-VOD and VWC for all three products was consistently high at 0.94, 0.96, and 0.86, respectively. The correlations with LAI were slightly lower, with R values of 0.76, 0.78, and 0.74, respectively. L-VOD is sensitive to the entire vegetation layer, including woody plants, which explains its higher spatial correlation with VWC and LAI. On the other hand, NDVI, being more sensitive to upper canopy vegetation characteristics, exhibited a lower spatial correlation with L-VOD, with values of 0.74, 0.79, and 0.70, respectively.

This finding aligns with the results of Li et al. (2018) [53]. Overall, the spatial gradients of VWC, LAI, and NDVI were well predicted by all three L-VOD products, with SMAP SCA-V demonstrating the highest correlation, followed by SMAP DCA and SMAP-IB.

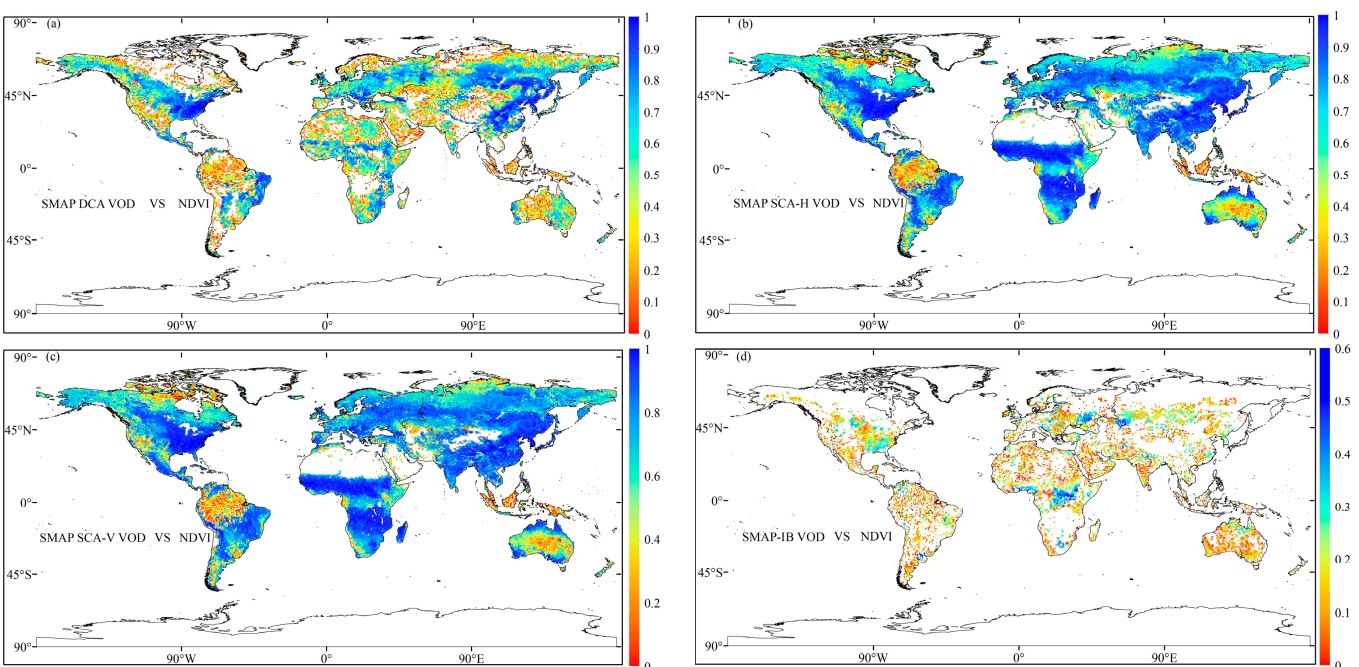

**Figure 13.** Correlation coefficients between different L-VOD products and NDVI over time. (**a**) SMAP DCA VOD, (**b**) SMAP SCA-H VOD, (**c**) SMAP SCA-V VOD and (**d**) SMAP-IB VOD.

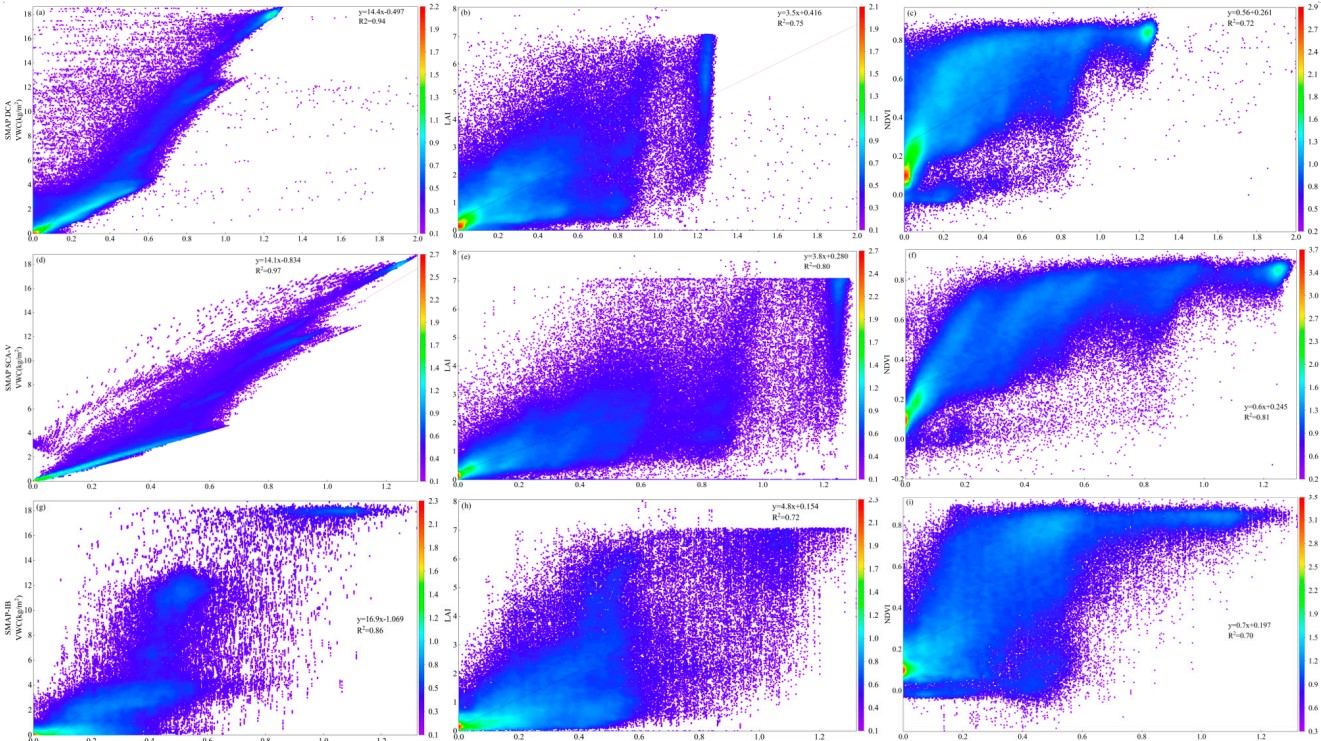

**Figure 14.** Density distribution maps at a global scale of three L-VOD products with VWC (first column), LAI (second column), and NDVI (last column). (**a**–**c**) SMAP DCA VOD, (**d**–**f**) SMAP SCA-V VOD and (**g**–**i**) SMAP-IB VOD.

The density distributions of SMAP DCA, SMAP SCA-V, and SMAP-IB L-VOD products vary globally. The SMAP DCA and SMAP SCA-V algorithms primarily retrieved L-VOD values ranging from 0 to 0.9, based on the distribution of L-VOD values. On the other hand, the SMAP-IB algorithm retrieved L-VOD values primarily between 0 and 0.6. Additionally, when L-VOD exceeded 0.6, the densities of SMAP DCA and SMAP-SCA-V were higher compared to that of SMAP-IB. At L-VOD values of 0.2, the corresponding densities of VWC, LAI, and NDVI ranged from 0 to 2, 0 to 1, and 0 to 0.2, respectively. At this point, the densities of L-VOD, VWC, LAI, and NDVI were the highest, indicating concentrated data. As the L-VOD values increased, the density decreased, and the data became more fragmented.

## 5. Conclusions

To enhance the application of remote sensing/reanalysis of SM products, this study conducted a comprehensive evaluation and comparison of 10 SM products (SMAP DCA, SMAP SCA-H, SMAP SCA-V, SMAP-IB, SMOS IC, SMOS L3, LPRM_C1, LPRM_C2, LPRM_X, and ERA5-Land) at a global scale, considering different spatial and environmental scenarios. The evaluation was performed using ISMN data from 2015 to 2022. The main conclusions are as follows:

(1) SMAP DCA and ERA5-Land outperformed other products in capturing SM changes across various climate zones, while the performance difference between LPRM_C1, LPRM_C2, and LPRM_X was minimal.

(2) The L-band products, including SMAP DCA, SMAP SCA-H, SMAP SCA-V, SMAP-IB, SMOS IC, and SMOS L3, exhibited high accuracy across diverse land cover types. However, LPRM_C1, LPRM_C2, and LPRM_X generally overestimated SM, whereas ERA5-Land outperformed both L-band and C/X-band products based on performance indicators.

(3) SMAP DCA, SMAP SCA-V, SMAP-IB, and ERA5-Land demonstrated strong SM inversion capabilities across all soil types. LPRM_C2 performed better in sandy soil (ARENOSOLS), while LPRM_X showed better performance in clay pan soil (PLANOSOLS). LPRM_X also exhibited superior performance in gray soil (PODZOLS) and volcanic ash soil (ANDOSOLS).

(4) SMAP-IB demonstrated better performance when analyzing dynamic factors under climatic conditions, displaying higher accuracy and providing a more accurate description of SM. SMAP DCA, SMAP SCA-V, and ERA5-Land outperformed SMOS IC and SMOS L3 across all evaluation indicators, while LPRM_C1, LPRM_C2, and LPRM_X exhibited larger errors during the SM retrieval process.

(5) From 2015 to 2022, significant regional variations in SM were observed. Wet tropical rainforests, dry temperate continental climates, and tropical desert climates exhibited distinct patterns. Wet coastal areas and dry inland areas showed contrasting SM levels, with SM increasing in warm-current coastal areas and decreasing in cold-current coastal areas. SM detected by SMAP SCA-H, SMOS L3, LPRM C1, LPRM C2, and LPRM X products exhibited a decreasing trend in large regions of the world, while other products showed moderate variations in most regions, with SM content remaining generally stable.

Although the performance of different SM products varied under dynamic and static conditions, these relationships are likely to be complex. The limited spatiotemporal data available in this study and the evaluation results may possess some contingencies. In the future, it is advisable to evaluate satellite remote sensing/reanalysis product data over a longer period, integrating land model SM products and precipitation data. The influence of factors such as cloud cover and surface roughness should be further considered alongside other influencing factors. Through assimilation algorithms and continuous iterative optimization inversion, this study successfully obtained stable and high-precision SM products, offering data references and theoretical support for the rational utilization of global water resources and the sustainable development of the ecological environment.

**Author Contributions:** Conceptualization, H.Y. and P.Z.; methodology, Y.G. and P.Z; software, Y.G. and P.Z.; validation, H.Y. and Q.Z.; formal analysis, H.Y. and Q.Z.; investigation, Y.G. and P.Z; resources, H.Y. and Q.Z.; data curation, Y.G. and P.Z; writing—original draft preparation, H.Y. and P.Z.; writing—review and editing, H.Y. and Q.Z.; visualization, H.Y. and P.Z.; supervision, H.Y. and Q.Z.; project administration, H.Y. and Q.Z.; funding acquisition, H.Y. and Q.Z. All authors have read and agreed to the published version of the manuscript.

**Funding:** This research was funded by the National Natural Science Foundation of China, grant number: 41961052;41961144019; the Inner Mongolia Natural Science Foundation Project, grant number: 2021MS04015; and the National Natural Science Foundation of China, grant number: 41661009.

**Institutional Review Board Statement:** Not applicable.

**Informed Consent Statement:** Not applicable.

**Data Availability Statement:** SMAP L3 data was sourced from https://nsidc.org/data/smap/data, accessed on 20 July 2022. SMAP-IB data was sourced from https://ib.remote-sensing.inrae.fr/, accessed on 22 July 2022. SMOS data was sourced from https://www.catds.fr/sipad/, accessed on 22 July 2022. LPRM dataset was sourced from https://gcmd.gsfc.nasa.gov/, accessed on 23 July 2022. ERA5-Land data was sourced from https://cds.climate.copernicus.eu/cdsapp#!/home, accessed on 23 July 2022. ISMN data was sourced from https://ismn.earth/en/, accessed on 24 July 2022.

**Acknowledgments:** The authors wish to thank SMAP, SMOS, LPRM, and ERA5-Land for providing satellite soil moisture products, and Vienna University of Technology for providing ISMN field data. We also appreciate all of the editors and reviewers for their valuable comments and suggestions to improve this manuscript.

**Conflicts of Interest:** The authors declare no conflict of interest.

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
