# Peer review of "Evaluation of Remote Sensing and Reanalysis Products for Global Soil Moisture Characteristics"

_sustainability, doi:10.3390/su15119112_

Round 1

Reviewer 1 Report

Firstly, I wish the authors have given the such a good attempt to map and assess the accuracy and applicability of land soil moisture products globally using SMAP DCA, 19 SMAP SCA-H, SMAP SCA-V, SMAP-IB, SMOS IC, SMOS L3, LPRM_C1, LPRM_C2, LPRM_X, and 20 ERA5. In this regard, authors conducted a wide-ranging confirmation of these remote sensing/reanalysis soil moisture products based on ISMN ground observation data under static conditions (climate zone, land cover type, and soil type) and dynamic conditions (soil moisture, leaf area index, and land surface temperature). In addition, the global soil moisture spatial and temporal distribution characteristics were analyzed, and vegetation effect on soil moisture products were discussed in this study. The author's findings provided a reference for improving the global scale satellite/reanalysis of soil moisture products and other soil moisture studies.

However, this research paper is not upright mean with soundness in technical and writing aspects, regarding this I have some major comments and suggestions which will be useful to update the paper.

Major comments:

1.      Language: in this research paper, language and sentence formations are poor, it is hard to understand some of the sentences (mentioned in minor corrections). Therefore, the following corrections should be undertaken by the authors to fix syntax and style, eliminate typos, and resolve several punctuation problems in the paper.

2.      Section 2. Materials and methods-

In this section, the authors should elaborately discuss section 2.4. Data processing, it seems concise in the paper, so that, it has to be improved.

3.      Figure 11 is missing in the paper.

4.      Section 4. The discussion part can be discussed in a concise manner. It is reflecting the results text in the discussion part as well.

5.      The conclusion part seems not upright, it has to be developed.

Minor comments

1.      Line No. 46 – Improper sentence

2.      Line No. 116- 119 - repeated sentence

3.      Line No. 122-125- poor sentence formation and rearranging the sentence

4.      Line No. 143-148 is not clear

5.      Line No. 381-382-  rearrange the sentence

6.      Line No.595- repeated word

7.      Line No.473- Controlled of South America?

8.      Line No.641-643- Sentence formation is poor

9.      Line No.667-669- Rearrange the sentence

10.  Line No.705- Reverse trend? Not clear.

In this research paper, language and sentence formations are poor, it is hard to understand some of the sentences (mentioned in minor corrections). Therefore, the following corrections should be undertaken by the authors to fix syntax and style, eliminate typos, and resolve several punctuation problems in the paper

Reviewer 2 Report

Soil moisture (SM) plays an important role in land surface energy and water cycles, affecting the interchange of energy, water, and carbon between the land and atmosphere. Therefore, obtaining accurate SM data is critical for drought monitoring, climate modeling, crop yield estimation, flood prediction, heat wave forecasting, and water resource management. Several rigorous validation studies on 10 SM products, including  SMAP DCA, SMAP SCA-H, SMAP SCA-V, SMAP-IB, SMOS IC, SMOS L3, land parameter retrieval models (LPRM_C1, LPRM_C2, and LPRM_X), and ERA5-land products were conducted. It can provide a reference for improving satellite/reanalysis of soil moisture products and other soil moisture studies on a global scale.

Moderate editing of English language

Reviewer 3 Report

This is an interesting and scientific paper. the author nicely explain and conducted a comprehensive verification of 10 remote sensing/reanalysis soil moisture products based on ISMN ground observation data under static conditions and dynamic conditions. In this paper global soil moisture spatial and temporal distribution characteristics were analysed, and vegetation effect on soil moisture products was discussed. The findings of the study provide a reference for improving satellite/reanalysis of soil moisture products and other soil moisture studies on a global scale. From this point of view, this paper has great importance in recent times. However, there are some observations that need to be adequately addressed (see below).

·         In Figure 1, please change the colour of the location of validation sites. It will be better for the visualization of points.

·         Lines 128 to 132, add references.

·         Lines 140-141, add the reference.

·         Line 151, add the reference.

·         In the material and methods section, add some essential characteristics of   10 SM products so that readers can understand about the products.

·         In the material and methods section, how LST was measured/from which database it was collected. Mentioned, please. Noting was mentioned about NDVI, need to add data source/method involved in NDVI. Also, add the source of soil type data and land cover type data, data characteristics and processing steps in the respective sections.

·         There is no figure added for Figure 11. May be missing.

·         Line 556, suddenly NDVI came up. So need some explanation/logic/data characteristics in the method section.

·         Be sure that when you first time used any abbreviation, please use the full form of that abbreviation. Because in this paper so many abbreviations are used.

·         In the conclusion section please add the importance/significance of the study's findings and future directions based on these findings.

Round 2

Reviewer 1 Report

All the revisions have been included in the revised version of manuscript. Therefore this manuscript can be accepted for the publication.